# BiLoRA: A Bi-level Optimization Framework for Low-Rank Adapters

## Abstract

Low-rank adaptations (LoRA) are widely employed for fine-tuning large-scale pretrained models in downstream tasks, by learning low-rank incremental matrices. LoRA and its variants such as AdaLoRA train an entire low-rank incremental matrix on a single training dataset, which often leads to overfitting to training data and inferior generalization on test data. To address this problem, we propose a bi-level optimization (BLO) based method for alleviating overfitting. Our method parameterizes a low-rank incremental matrix in a pseudo singular value decomposition form, and separates the training of pseudo singular vectors and values onto different data subsets in different optimization problems. This separation alleviates the risk of overfitting to a single dataset and improves generalization on other data. Specifically, in the lower level of our BLO formulation, we train the pseudo singular vectors on a subset of the training data. In the upper level, we learn the pseudo singular values on the other subset of the training data. The two levels of optimization problems are mutually dependent on each other and solved jointly. On ten datasets from natural language understanding and generation tasks and on various popular large pretrained models, our method achieves significantly better performance than LoRA, AdaLoRA, and other fine-tuning baseline methods with similar amounts of trainable parameters.

## 1 Introduction

Large language models (LLMs) have achieved excellent performance across various natural language processing tasks (Devlin et al., 2018; He et al., 2020; Radford et al., 2019; Brown et al., 2020). The prevalent paradigm for leveraging large language models in application development involves pretraining on large-scale data and subsequently fine-tuning the pretrained model on specific downstream tasks. With the ever-increasing size of large language models, full fine-tuning (Qiu et al., 2020) them on various downstream tasks can cause significant computation costs. In addition, the large number of parameters in pre-trained models may make the fine-tuning process more prone to overfitting (Karimi Mahabadi et al., 2021). Researchers have proposed multiple fine-tuning methods to address these issues. These methods, aiming to reduce the parameter count during fine-tuning while maintaining performance, can be collectively referred to as Parameter-Efficient Fine-Tuning (PEFT) methods (Houlsby et al., 2019; Ding et al., 2023; Mao et al., 2021).

Low-Rank Adaptation (LoRA) (Hu et al., 2021) is one of the important methods for PEFT. Different from adapter tuning (Houlsby et al., 2019; Rebuffi et al., 2017; Pfeiffer et al., 2020), LoRA does not add small neural modules to the pre-trained model. LoRA takes inspiration from Li et al. (2018); Aghajanyan et al. (2020) which show that well trained over-parameterized models actually exist within a space characterized by a low intrinsic dimension. It introduces incremental updates named low-rank adapters to frozen pre-trained weights and parameterizes them in the form of the product of two much smaller matrices. For $h = W_0 x$, the modified forward pass yields: $h = W_0 x + \Delta W x = W_0 x + BAx$, where $\Delta W \in \mathbb{R}^{d \times k}$, $A \in \mathbb{R}^{d \times r}$, $B \in \mathbb{R}^{r \times k}$ and $r \ll min\{d, k\}$. With much less trainable parameters, LoRA achieves comparable or even better performance than full fine-tuning and other adaptation methods (Hu et al., 2021).

LoRA sets the rank of incremental matrices at different layers to be the same, without considering the fact that pretrained weight matrices in different layers have varying importance for a downstream task. A more important weight matrix should be finetuned more, with a larger number of weight

parameters (equivalently, a larger rank) in its incremental matrix. To address this issue, AdaLoRA (Zhang et al., 2023) sets different ranks for incremental matrices at different layers adaptively according to layers' importance. It parameterizes a low-rank incremental matrix $\Delta W$ as $\Delta W = P\Lambda Q$ to mimic SVD. With regularization to enforce the orthogonality of $P$ and $Q$, $\Lambda$ can be approximately considered as a singular value matrix. AdaLoRA uses singular values and vectors to compute important scores for determining how to set layer-specific ranks.

One limitation of AdaLoRA is that it learns pseudo singular vectors in $\{P, Q\}$ and pseudo singular values in $\Lambda$ simultaneously by minimizing the fine-tuning loss on a single training dataset. This often results in overfitting to the training data and unsatisfactory generalization on test data. Particularly, $\Lambda$ determines the number of learnable parameters and the contribution of each rank-1 update matrix (outer product of two pseudo singular vectors) in $\Delta W$. Learning $\Lambda$ by minimizing a single dataset's training loss can easily render these contributions and parameter amounts tailored to this dataset, leading to inferior generalization performance on other data.

To address this problem, we propose a bi-level optimization (BLO) based method to learn $\{P, Q\}$ and $\Lambda$ on different subsets of the training data. A BLO formulation (Sinha et al., 2017) consists of two levels of nested optimization problems. The optimal variables in the lower level are the inputs of the objective function in the upper level. The non-optimal variables in the upper level are the inputs of the objective function in the lower level. In the lower level of our formulation, we train $\{P, Q\}$ by minimizing a fine-tuning loss on a subset $S$ of the training dataset $D$ while tentatively fixing $\Lambda$. The optimally learned $\{P^*(\Lambda), Q^*(\Lambda)\}$ are functionals of $\Lambda$. In the upper level, we validate $\{P^*(\Lambda), Q^*(\Lambda)\}$ on the rest of the training data $D\backslash S$. The validation loss is a function of $\Lambda$ and we learn $\Lambda$ by minimizing this loss. By separating the learning of $\{P, Q\}$ and $\Lambda$ onto different data subsets in different optimization problems, our method can effectively alleviate overfitting to a single dataset and improve generalization performance to other datasets.

Our contributions can be summarized as follows:

- We propose a novel bi-level optimization based method to alleviate overfitting in LoRA and its variants. Different from previous methods which learn an entire incremental matrix on a single dataset, our method separates the learning of parameter subsets onto different datasets in different optimization problems which are tightly coupled. In this way, our method can effectively alleviate overfitting to a single dataset.

- We demonstrate the effectiveness of our method on ten datasets in both natural language understanding and generation tasks and on various pretrained large models including RoBERTa, DeBERTa, and GPT2. Compared with LoRA, AdaLoRA and other popular fine-tuning methods, our method achieves significantly better performance with similar amounts of trainable parameters.

## 2 RELATED WORK

**Low-Rank Adaptation.** Li et al. (2018) and Aghajanyan et al. (2020) demonstrate that widely-used pre-trained models possess a very low intrinsic dimension and it is possible to achieve comparable fine-tuning performance by utilizing a reparameterization with reduced dimensionality. This inspires low-rank adapters to be introduced for fine-tuning. LoRA introduces incremental updates to frozen pre-trained weights as low-rank adapters (Hu et al., 2021). By parameterizing the low-rank adapter as the product of two low-rank matrices, LoRA greatly reduces trainable parameters while maintaining or even improving the performance over full fine-tuning. Multiple methods have been proposed to improve the time/memory efficiency and performance of low-rank adapters based on LoRA. DyLoRA (Valipour et al., 2022) trains low-rank adapter blocks for multiple ranks by sorting the learned representations dynamically during training. QLoRA (Dettmers et al., 2023) introduces multiple strategies to reduce memory footprint for low-rank adapters, lowering the memory barrier for training LLMs. LoraHub (Huang et al., 2023) is designed to facilitate the efficient combination of LoRA modules trained on various tasks using only a few examples from a new task. AdaLoRA (Zhang et al., 2023) allocates the parameter budget adaptively according to the importance of modules to improve the fine-tuning performance in specific budget settings. It parameterizes the incremental updates in the form of singular value decomposition and iteratively prunes singular values in correspondence to their importance scores during training. Different from these existing methods

which train all the parameters in incremental updates on a single training dataset and therefore often lead to overfitting, our method (based on the SVD reparameterization of incremental updates) separately train singular values and singular vectors in two different optimization levels, which effectively alleviates the risk of overfitting to a single dataset.

**Bi-level Optimization (BLO).** BLO has gained much attention for formulating various machine learning methods including meta-learning (Finn et al., 2017; Rajeswaran et al., 2019), hyperparameter optimization (Franceschi et al., 2017; Lorraine et al., 2020), neural architecture search (Liu et al., 2018; Zhang et al., 2021), reinforcement learning (Rajeswaran et al., 2020), to name a few. In addition to applying BLO to various machine learning problems, various algorithms have been proposed to address this specific form of optimization problem, including zeroth-order methods like Bayesian optimization (Cui & Bai, 2019), first-order algorithms based on hypergradients (Pearlmutter & Siskind, 2008; Lorraine et al., 2020), etc. Gradient-based BLO is efficient for scaling up to high-dimensional problems with a large number of trainable parameters. We expand the application scenarios of gradient-based BLO and build an efficient training framework to improve the generalization performance of low-rank adapters.

## 3 METHODS

We propose BiLoRA (Figure 1), a novel low-rank adapter training framework based on bi-level optimization. Similar to AdaLoRA, incremental matrices in our method are parameterized in a pseudo SVD form with learnable pseudo singular vectors $\mathcal{V}$ and pseudo singular values $\mathcal{E}$. We split the training dataset into two non-overlapping subsets $D_1$ and $D_2$. In the lower level, we train $\mathcal{V}$ on $D_1$ while fixing $\mathcal{E}$. The optimal solution $\mathcal{V}^*(\mathcal{E})$ (which is a functional of $\mathcal{E}$) is fed into the upper level. In the upper level, we train $\mathcal{E}$ on the dataset $D_2$. The updated $\mathcal{E}$ is fed into the lower level. The two levels of optimization problems are solved iteratively until convergence.

### 3.1 PARAMETERIZATION OF LOW-RANK INCREMENTAL MATRICES

Following (Zhang et al., 2023), we parameterize the low-rank incremental matrix $\Delta W$ as $\Delta W = P\Lambda Q$ which mimics SVD. The diagonal matrix $\Lambda$ contains

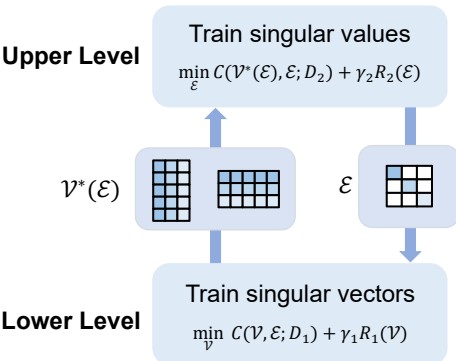

**Bi-level Optimization Framework**

Upper Level — Train singular values
$$\min_{\mathcal{E}} C(\mathcal{V}^*(\mathcal{E}), \mathcal{E}; D_2) + \gamma_2 R_2(\mathcal{E})$$

$\mathcal{V}^*(\mathcal{E})$     $\mathcal{E}$

Lower Level — Train singular vectors
$$\min_{\mathcal{V}} C(\mathcal{V}, \mathcal{E}; D_1) + \gamma_1 R_1(\mathcal{V})$$

Figure 1: The proposed BiLoRA method.

*pseudo singular values* and the approximately orthogonal matrices $P$ and $Q$ represent *pseudo left/right singular vectors*. We use $k$ to index the incremental matrix, i.e., $\Delta W_k = P_k \Lambda_k Q_k$ for $k = 1, ..., n$, where n is the number of low-rank adapters. We denote the $i$-th singular value of $\Delta W_k$ as $\lambda_{k,i}$ and the rank of low-rank adapters as $r$. We further denote the parameter sets as $\mathcal{P} = \{P_k\}_{k=1}^n$, $\mathcal{E} = \{\Lambda_k\}_{k=1}^n$, $\mathcal{Q} = \{Q_k\}_{k=1}^n$, and $\mathcal{V} = \{\mathcal{P}, \mathcal{Q}\}$. To encourage $P_k$ and $Q_k$ to be approximately orthogonal, we use the following regularizer as in AdaLoRA (Zhang et al., 2023):

$$R_1 = \sum_{k=1}^n (\|P_k^T P_k - I\|_F^2 + \|Q_k Q_k^T - I\|_F^2), \tag{1}$$

where $I$ is an identity matrix and $\| \cdot \|_F$ denotes the Frobenius norm.

**Parameterization of Pseudo Singular Values.** We parameterize the pseudo singular values in $\Lambda$ in three specific forms.

- **Real-Value**: All pseudo singular values are real-valued without any constraints.
- **Softmax**: Given a real vector $v$, we apply the softmax operation to it. $softmax(v)$ are used as the pseudo singular values. These values add up to one and represent the contributions of their corresponding singular vector pairs.

- **Approximately Binary**: Given a real vector $v$, we apply element-wise sigmoid to it to transform the values in $v$ into $(0, 1)$. Then we use an element-wise entropy regularizer to encourage the values in $sigmoid(v)$ are close to either zero or one. The regularizer is defined as:

$$R_2(\mathcal{E}) = \sum_{k=1}^{n} \sum_{i=1}^{r} \lambda_{k,i} \log \lambda_{k,i} + (1 - \lambda_{k,i}) \log(1 - \lambda_{k,i}). \tag{2}$$

This setting automatically assigns either a high or low importance to each singular vector pair with the corresponding singular value as zero or one, effectively serving as an automatic rank selection mechanism.

## 3.2 A Bi-level Optimization Framework

Our method is based on bi-level optimization, where pseudo singular vector matrices $\mathcal{V}$ and their corresponding pseudo singular value matrices $\mathcal{E}$ are set as trainable parameters for the lower and upper level respectively.

**Lower Level.** In the lower level, we perform LoRA fine-tuning of a pre-trained model by minimizing a loss $C$ defined on the first dataset $D_1$ and low-rank incremental matrices $\{\Delta W_k\}_{k=1}^{n}$. Calculating $C$ involves the forward pass for each input example $x$: $W_0 x + \Delta W x = W_0 x + P \Lambda Q x$, where $W_0$ is a weight matrix in the pretrained model. $R_1$ in Eq.(1) is applied to promote the approximate orthogonality of $P$ and $Q$. The overall training objective is $L_1 = C(\mathcal{V}, \mathcal{E}; D_1) + \gamma_1 R_1(\mathcal{V})$, where $\gamma_1$ is a tradeoff parameter. In this level, we only train $\mathcal{V}$, while keeping $\mathcal{E}$ tentatively fixed. $\mathcal{E}$ will be updated in the upper level. In the end, the lower level amounts to solving the following problem:

$$\mathcal{V}^*(\mathcal{E}) = \arg\min_{\mathcal{V}} \ C(\mathcal{V}, \mathcal{E}; D_1) + \gamma_1 R_1(\mathcal{V}). \tag{3}$$

$\mathcal{V}^*(\mathcal{E})$ denotes that the optimal solution $\mathcal{V}^*$ depends on $\mathcal{E}$ since $\mathcal{V}^*$ depends on $C$ which depends on $\mathcal{E}$.

**Upper Level.** In the upper level, we validate the fine-tuned model where the incremental matrices are parameterized by the optimally learned $\mathcal{V}^*(\mathcal{E})$ and unlearned pseudo singular values in $\mathcal{E}$, on the second dataset $D_2$. This results in a validation loss $C(\mathcal{V}^*(\mathcal{E}), \mathcal{E}, D_2)$, which is a function of $\mathcal{E}$. We learn $\mathcal{E}$ by minimizing this loss. Optionally, we use the regularizer $R_2$ in Eq.(2) to encourage the pseudo singular values in $\mathcal{E}$ to be approximately binary. The overall objective function is $L_2 = C(\mathcal{V}^*(\mathcal{E}), \mathcal{E}; D_2) + \gamma_2 R_2(\mathcal{E})$, where $\gamma_2$ is a tradeoff parameter. This level amounts to solving the following optimization problem:

$$\min_{\mathcal{E}} \ C(\mathcal{V}^*(\mathcal{E}), \mathcal{E}; D_2) + \gamma_2 R_2(\mathcal{E}). \tag{4}$$

**A Bi-level Optimization Framework.**

Integrating these two interdependent levels of optimization problems, we have the following bi-level optimization framework:

$$\text{Upper Level:} \qquad \min_{\mathcal{E}} \ C(\mathcal{V}^*(\mathcal{E}), \mathcal{E}; D_2) + \gamma_2 R_2(\mathcal{E})$$

$$\text{Lower Level:} \qquad s.t. \ \mathcal{V}^*(\mathcal{E}) = \arg\min_{\mathcal{V}} \ C(\mathcal{V}, \mathcal{E}; D_1) + \gamma_1 R_1(\mathcal{V})$$

Note that these two levels of optimization problems are mutually dependent on each other. The output of the lower level, which is $\mathcal{V}^*(\mathcal{E})$, is the input of the upper level. The optimization variable $\mathcal{E}$ in the upper level is the input of the lower level. By solving these two interconnected problems jointly, we can learn the pseudo singular vectors and values end-to-end.

**Optimization Algorithm.** We utilize a gradient-based optimization algorithm (Choe et al., 2022) to solve this bi-level optimization problem. Our overall optimization algorithm is summarized in Algorithm 1. Specifically, in the lower level, we perform gradient descent for a preset number of steps $T_1$ on the pseudo singular vector matrices $\mathcal{V}$ to approximate the optimal solution $\mathcal{V}^*(\mathcal{E})$. With the initial $\mathcal{V}$ as $\mathcal{V}^{(0)}$ and learning rate $\eta_1$, the gradient descent steps can be formulated as:

$$\mathcal{V}^{(t)} = \mathcal{V}^{(t-1)} - \eta_1 \frac{dL_1}{d\mathcal{V}^{(t-1)}}, \text{ for } t = 1, 2, 3, ..., T_1.$$

Table 1: RoBERTa$_{\text{base/large}}$ (R$_{\text{b/l}}$) with different adaptation methods on the GLUE benchmark. We report the average result of five runs with different random seeds. Higher is better for all metrics. Numbers except BiLoRA are published in prior works. $*$ indicates model already adapted to MNLI when adapting to MRPC, RTE, and STS-B, while † indicates model started as pre-trained when adapting to all datasets.

| Method | Params | MNLI | SST-2 | MRPC | CoLA | QNLI | QQP | RTE | STS-B | Avg. |
|---|---|---|---|---|---|---|---|---|---|---|
| R$_b$(FT) | 125.0M | 87.6 | 94.8 | 90.2 | 63.6 | 92.8 | **91.9** | 78.7 | 91.2 | 86.4 |
| R$_b$(BitFit) | 0.1M | 84.7 | 93.7 | **92.7** | 62.0 | 91.8 | 84.0 | 81.5 | 90.8 | 85.2 |
| R$_b$(Adpt$^D$) | 0.3M | 87.1$_{\pm.0}$ | 94.2$_{\pm.1}$ | 88.5$_{\pm1.1}$ | 60.8$_{\pm.4}$ | 93.1$_{\pm.1}$ | 90.2$_{\pm.0}$ | 71.5$_{\pm2.7}$ | 89.7$_{\pm.3}$ | 84.4 |
| R$_b$(Adpt$^D$) | 0.9M | 87.3$_{\pm.1}$ | 94.7$_{\pm.3}$ | 88.4$_{\pm.1}$ | 62.6$_{\pm.9}$ | 93.0$_{\pm.2}$ | 90.6$_{\pm.0}$ | 75.9$_{\pm2.2}$ | 90.3$_{\pm.1}$ | 85.4 |
| R$_b$(LoRA)$^*$ | 0.3M | 87.5$_{\pm.3}$ | **95.1$_{\pm.2}$** | 89.7$_{\pm.7}$ | 63.4$_{\pm1.2}$ | **93.3$_{\pm.3}$** | 90.8$_{\pm.1}$ | 86.6$_{\pm.7}$ | 91.5$_{\pm.2}$ | 87.2 |
| R$_b$(BiLoRA)$^*$ | 0.3M | **87.9$_{\pm.2}$** | **95.1$_{\pm.2}$** | 91.7$_{\pm.5}$ | **64.8$_{\pm.6}$** | **93.3$_{\pm.2}$** | 91.4$_{\pm.2}$ | **87.2$_{\pm.4}$** | **91.7$_{\pm.2}$** | **87.9** |
| R$_l$(FT)$^*$ | 355.0M | 90.2 | 96.4 | 90.9 | 68.0 | 94.7 | **92.2** | 86.6 | 92.4 | 88.9 |
| R$_l$(LoRA)$^*$ | 0.8M | **90.6$_{\pm.2}$** | 96.2$_{\pm.5}$ | 90.9$_{\pm1.2}$ | 68.2$_{\pm1.9}$ | 94.9$_{\pm.3}$ | 91.6$_{\pm.1}$ | 87.4$_{\pm2.5}$ | **92.6$_{\pm.2}$** | 89.0 |
| R$_l$(BiLoRA)$^*$ | 0.8M | **90.6$_{\pm.3}$** | **96.7$_{\pm.4}$** | **92.6$_{\pm1.4}$** | **69.2$_{\pm1.6}$** | **95.0$_{\pm.1}$** | 92.0$_{\pm.1}$ | **89.5$_{\pm1.1}$** | **92.6$_{\pm.8}$** | **89.8** |
| R$_l$(Adpt$^P$)$^†$ | 3.0M | 90.2$_{\pm.3}$ | 96.1$_{\pm.3}$ | 90.2$_{\pm.7}$ | 68.3$_{\pm1.0}$ | 94.8$_{\pm.2}$ | 91.9$_{\pm.1}$ | 83.8$_{\pm2.9}$ | 92.1$_{\pm.7}$ | 88.4 |
| R$_l$(Adpt$^P$)$^†$ | 0.8M | 90.5$_{\pm.3}$ | 96.6$_{\pm.2}$ | 89.7$_{\pm1.2}$ | 67.8$_{\pm2.5}$ | 94.8$_{\pm.3}$ | 91.7$_{\pm.2}$ | 80.1$_{\pm2.9}$ | 91.9$_{\pm.4}$ | 87.9 |
| R$_l$(Adpt$^H$)$^†$ | 6.0M | 89.9$_{\pm.5}$ | 96.2$_{\pm.3}$ | 88.7$_{\pm2.9}$ | 66.5$_{\pm4.4}$ | 94.7$_{\pm.2}$ | **92.1$_{\pm.1}$** | 83.4$_{\pm1.1}$ | 91.0$_{\pm1.7}$ | 87.8 |
| R$_l$(Adpt$^H$)$^†$ | 0.8M | 90.3$_{\pm.3}$ | 96.3$_{\pm.5}$ | 87.7$_{\pm1.7}$ | 66.3$_{\pm2.0}$ | 94.7$_{\pm.2}$ | 91.5$_{\pm.1}$ | 72.9$_{\pm2.9}$ | 91.5$_{\pm.5}$ | 86.4 |
| R$_l$(LoRA)$^†$ | 0.8M | **90.6$_{\pm.2}$** | 96.2$_{\pm.5}$ | 90.2$_{\pm1.0}$ | 68.2$_{\pm1.9}$ | 94.8$_{\pm.3}$ | 91.6$_{\pm.2}$ | 85.2$_{\pm1.1}$ | 92.3$_{\pm.5}$ | 88.6 |
| R$_l$(BiLoRA)$^†$ | 0.8M | **90.6$_{\pm.3}$** | **96.7$_{\pm.4}$** | **92.2$_{\pm1.0}$** | **69.2$_{\pm1.6}$** | **95.0$_{\pm.1}$** | 92.0$_{\pm.1}$ | 87.4$_{\pm1.0}$ | **92.6$_{\pm.8}$** | **89.5** |

We plug $\mathcal{V}^*(\mathcal{E}) \approx \mathcal{V}^{(T_1)}$ into the overall objective function in the upper level and get an approximate objective $\widehat{L}_2 = C(\mathcal{V}^{(T_1)}, \mathcal{E}; D_2) + \gamma_2 R_2(\mathcal{E})$. We perform gradient descent for a preset number of steps $T_2$ on the pseudo singular values in $\mathcal{E}$ to minimize $\widehat{L}_2$. With the initial $\mathcal{E}$ as $\mathcal{E}^{(0)}$ and learning rate $\eta_2$, the gradient descent steps can be formulated as:

$$\mathcal{E}^{(t)} = \mathcal{E}^{(t-1)} - \eta_2 \frac{d\widehat{L}_2}{d\mathcal{E}^{(t-1)}}, \text{ for } t = 1, 2, 3, ..., T_2.$$

These steps constitute one global optimization step. We take iterative global steps between the lower level and upper level to solve this bi-level optimization problem until converge. Specifically, following the chain rule, the hypergradient for the upper level can be calculated as:

$$\frac{d\widehat{L}_2}{d\mathcal{E}} = \frac{\partial \widehat{L}_2}{\partial \mathcal{E}} + \frac{\partial \mathcal{V}^{(T_1)}}{\partial \mathcal{E}} \times \frac{\partial \widehat{L}_2}{\partial \mathcal{V}^{(T_1)}}.$$

---

**Algorithm 1** BiLoRA

1: **Input**: Datasets $D_1$, $D_2$; unroll steps $T_1$, $T_2$; learning rates $\eta_1$, $\eta_2$.
2: **In a Global Step do**
3:      **for** $t = 1, 2, 3, ..., T_1$ **do**
4:          Sample a minibatch $B_1^{(t)}$ from $D_1$
5:          Compute $\frac{dL_1}{d\mathcal{V}^{(t-1)}}$ on $B_1^{(t)}$ and update $\mathcal{V}^{(t)} = \mathcal{V}^{(t-1)} - \eta_1 \frac{dL_1}{d\mathcal{V}^{(t-1)}}$
6:      **for** $t = 1, 2, 3, ..., T_2$ **do**
7:          Sample a minibatch $B_2^{(t)}$ from $D_2$
8:          Compute $\frac{d\widehat{L}_2}{d\mathcal{E}^{(t-1)}} = \frac{\partial \widehat{L}_2}{\partial \mathcal{E}^{(t-1)}} + \frac{\partial \mathcal{V}^{(T_1)}}{\partial \mathcal{E}^{(t-1)}} \times \frac{\partial \widehat{L}_2}{\partial \mathcal{V}^{(T_1)}}$ and update $\mathcal{E}^{(t)} = \mathcal{E}^{(t-1)} - \eta_2 \frac{d\widehat{L}_2}{d\mathcal{E}^{(t-1)}}$
9: **end this step**

---

## 4 EXPERIMENTS

We evaluated the downstream performance of BiLoRA on RoBERTa (Liu et al., 2019), DeBERTa (He et al., 2020) and GPT-2 (Radford et al., 2019), and compared with LoRA (Hu et al., 2021), AdaLoRA (Zhang et al., 2023), and other baselines. Our experiments covered a wide range of tasks, from natural language understanding (NLU) to generation (NLG). Specifically, we evaluated

Table 2: DeBERTa-v3-base ($D_{v3}$) with different adaptation methods, on the GLUE benchmark. We report the average result of five runs with different random seeds. Higher is better. * indicates numbers published in prior works. BiLoRA outperforms FT, LoRA, AdaLoRA, and other adaptation methods with equal or less parameters.

| Method | Params | MNLI | SST-2 | MRPC | CoLA | QNLI | QQP | RTE | STS-B | Avg. |
|---|---|---|---|---|---|---|---|---|---|---|
| $D_{v3}$(FT)* | 184.0M | 90.01 | 95.63 | 89.46 | 69.19 | 94.03 | 92.40 | 83.75 | 91.60 | 88.09 |
| $D_{v3}$(Adpt$^H$)* | 0.6M | 90.18 | 95.30 | 89.22 | 67.87 | 93.76 | 91.65 | 85.56 | 91.30 | 87.93 |
| $D_{v3}$(Adpt$^P$)* | 0.6M | 90.22 | 95.53 | 89.22 | 69.48 | 93.98 | 91.62 | 84.12 | 91.52 | 88.04 |
| $D_{v3}$(LoRA)* | 0.3M | 90.34 | 94.95 | 89.71 | 68.71 | 94.03 | 91.61 | 85.56 | 91.68 | 88.15 |
| $D_{v3}$(AdaLoRA)* | 0.3M | 90.68 | 95.80 | 90.44 | 70.04 | **94.49** | 91.78 | 87.36 | 91.63 | 88.86 |
| $D_{v3}$(BiLoRA) | 0.3M | **90.81** | **96.02** | **91.42** | **70.52** | 94.25 | **91.82** | **88.45** | **91.96** | **89.41** |

RoBERTa and DeBERTa on the GLUE benchmark (Wang et al., 2018) and GPT-2 on the E2E NLG challenge (Novikova et al., 2017). We used DeBERTa-xxlarge(1.5B) to evaluate the scaling-up performance of our method. We used NVIDIA A100 for all experiments.

## 4.1 BASELINES

We compared with the same baselines as LoRA and AdaLoRA, and used the reported results in previous work. Additionally, we also took LoRA and AdaLoRA as our baselines to evaluate the effectiveness of our method.

**Full Fine-Tuning (FT)** is a frequently employed method for adaptation. The model is initialized with pre-trained weights and biases and all model parameters are subjected to gradient updates. We also included a simple variant reported in prior work on GPT-2 (Li & Liang, 2021), which only adapts the last two layers while freezing others.

**Bias-only or BitFit** (Zaken et al., 2021) is an effective PEFT method which only trains the bias vectors while freezing everything else in the pre-trained model.

**Prefix-embedding tuning (PreEmbed)** introduces specialized tokens within the input tokens, featuring trainable word embeddings that typically do not belong to the model's vocabulary (Li & Liang, 2021).

**Prefix-layer tuning (PreLayer)** learns the activations after every Transformer layer by replacing the activations computed from previous layers with trainable parameters. This method can be seen as an extension to prefix-embedding tuning.

**Adapter tuning** (Houlsby et al., 2019) inserts layer-adapters between neural modules such as the MLP module or the self-attention module. We used four types of adapters as in LoRA (Hu et al., 2021): **Adapter$^L$** with the adapter layer applied only after the MLP module and after a LayerNorm (Lin et al., 2020), **Adapter$^D$** with some adapter layers dropped for increasing efficiency (Rücklé et al., 2020). **Adapter$^H$** incorporates two fully connected layers within an adapter layer, with non-linearity in between (Houlsby et al., 2019). **Adapter$^P$** (Pfeiffer et al., 2020) is similar to **Adapter$^L$**, but introduces a novel two-stage transfer learning strategy to combine the knowledge from multiple source tasks.

**LoRA** (Hu et al., 2021) adds trainable incremental update matrices to pretrained weight matrices. Following the experimental settings of LoRA, we applied BiLoRA to $W_q$ and $W_v$ matrices (the query and value weight matrices in the self-attention module) for a fair comparison.

**AdaLoRA** (Zhang et al., 2023) proposes SVD-based adaptation and rank-allocation based on LoRA, which formulates the incremental matrices in the form of singular value decomposition and allocates rank budget based on importance scores.

## 4.2 NATURAL LANGUAGE UNDERSTANDING

For natural language understanding (NLU) tasks, we conducted experiments on the General Language Understanding Evaluation (GLUE) benchmark for RoBERTa and DeBERTa. Please see Appendix A for more details on the models and datasets we use.

Table 3: GPT-2 medium (M) and large (L) with different adaptation methods on the E2E NLG Challenge. For all metrics, higher is better. * indicates numbers published in prior works. We keep the same experimental settings as different adaptation baselines for a fair comparison.

| Model&Method | Params | BLEU | NIST | MET | ROUGE-L | CIDEr |
|---|---|---|---|---|---|---|
| GPT-2 M(FT)* | 354.92M | 68.2 | 8.62 | 46.2 | 71.0 | 2.47 |
| GPT-2 M(Adpt$^L$)* | 0.37M | 66.3 | 8.41 | 45.0 | 69.8 | 2.40 |
| GPT-2 M(Adpt$^L$)* | 11.09M | 68.9 | 8.71 | 46.1 | 71.3 | 2.47 |
| GPT-2 M(Adpt$^H$)* | 11.09M | $67.3_{\pm.6}$ | $8.50_{\pm.07}$ | $46.0_{\pm.2}$ | $70.7_{\pm.2}$ | $2.44_{\pm.01}$ |
| GPT-2 M(FT$^{Top2}$)* | 25.19M | 68.1 | 8.59 | 46.0 | 70.8 | 2.41 |
| GPT-2 M(PreLayer)* | 0.35M | 69.7 | 8.81 | 46.1 | 71.4 | 2.49 |
| GPT-2 M(LoRA)* | 0.35M | $70.4_{\pm.1}$ | $8.85_{\pm.02}$ | $46.8_{\pm.2}$ | $71.8_{\pm.1}$ | $2.53_{\pm.02}$ |
| GPT-2 M(BiLoRA) | 0.35M | $\mathbf{70.5}_{\pm.4}$ | $\mathbf{8.86}_{\pm.03}$ | $\mathbf{46.9}_{\pm.1}$ | $\mathbf{72.0}_{\pm.2}$ | $\mathbf{2.54}_{\pm.03}$ |
| GPT-2 L(FT)* | 774.03M | 68.5 | 8.78 | 46.0 | 69.9 | 2.45 |
| GPT-2 L(Adpt$^L$)* | 0.88M | $69.1_{\pm.1}$ | $8.68_{\pm.03}$ | $46.3_{\pm.0}$ | $71.4_{\pm.2}$ | $\mathbf{2.49}_{\pm.0}$ |
| GPT-2 L(Adpt$^L$)* | 23.00M | $68.9_{\pm.3}$ | $8.70_{\pm.04}$ | $46.1_{\pm.1}$ | $71.3_{\pm.2}$ | $2.45_{\pm.02}$ |
| GPT-2 L(PreLayer)* | 0.77M | 70.3 | 8.85 | 46.2 | 71.7 | 2.47 |
| GPT-2 L(LoRA)* | 0.77M | $70.4_{\pm.1}$ | $8.89_{\pm.02}$ | $46.8_{\pm.2}$ | $\mathbf{72.0}_{\pm.2}$ | $2.47_{\pm.02}$ |
| GPT-2 L(BiLoRA) | 0.77M | $\mathbf{70.5}_{\pm.3}$ | $\mathbf{8.90}_{\pm.04}$ | $\mathbf{47.0}_{\pm.3}$ | $\mathbf{72.0}_{\pm.4}$ | $2.49_{\pm.03}$ |

**Implementation Details.** Our implementation is based on *Huggingface Transformers* (Wolf et al., 2019) and *Betty* (Choe et al., 2022). *Betty* is a software library for solving large-scale multilevel optimization (MLO) problems. Specifically, we load RoBERETa and DeBERTa models with *Huggingface Transformers* and build our bi-level optimization framework with *Betty*.

**Experimental Settings.** Following LoRA, we used the development set in GLUE as test data since the test set is not publicly available. We divided the training set into two datasets, with an 8:2 split, serving as the lower-level and upper-level datasets respectively in our bi-level formulation. We maintained this fixed ratio for all tasks. Singular values were parameterized as Softmax if not otherwise stated and $R_1$ was added to the lower level as a regularizer. For RoBERTa base/large, we kept our experimental settings the same as LoRA. For DeBERTa-v3-base, we kept our experimental settings close to AdaLoRA while maintaining a lower parameter budget. We also kept hyperparameters such as sequence length, total batch size, LoRA rank, and LoRA alpha exactly the same as LoRA/AdaLoRA where necessary. These experimental settings allow for a fair comparison with all baseline methods. Please see the Appendix for all the hyperparameter settings.

**Main Results.** The same as LoRA, we report the overall (matched and mismatched) accuracy for MNLI, Matthew's correlation for CoLA, Pearson correlation for STS-B, and accuracy for the other tasks. Table 1 shows the results of RoBERTa base/large on the GLUE development set. As can be seen, our method outperforms LoRA on all datasets with the same number of trainable parameters. On most datasets, our method achieves better or on par performance compared with baselines. The average score of BiLoRA notably outperforms all the baselines. Table 2 shows the results of DeBERTa-v3-base on the GLUE development set. BiLoRA outperforms all baselines with equal or less trainable parameters. The improvements achieved by our method over baselines are attributed to its bi-level learning mechanism which separates the training of pseudo singular vectors and values on two distinct datasets. As a result, it effectively alleviates the risk of overfitting to one dataset and yields better generalization performance. In contrast, baseline methods train all parameters on the same dataset and thus lead to overfitting to this dataset. This is particularly evidenced by the observation that on smaller datasets such as CoLA, RTE, and MRPC where overfitting is more likely to occur, BiLoRA outperforms baselines by a larger margin.

## 4.3 NATURAL LANGUAGE GENERATION

For natural language generation (NLG) tasks, we followed the setup of Prefix-Tuning (Li & Liang, 2021) and LoRA (Hu et al., 2021) on GPT-2 for a direct comparison with LoRA and other adaptation methods. We evaluated GPT-2 medium and large on the E2E NLG Challenge. Please see Appendix A for more details on the models and datasets we used.

**Implementation Details.** Our implementation is based on the fine-tuning code for GPT-2 in Huggingface and Betty (Choe et al., 2022). Specifically, we load GPT-2 models with the code of Huggingface and build our bi-level optimization framework with Betty.

**Experimental Settings.** In our method, the training set and validation set are used as the lower-level and upper-level datasets respectively, and we report performance on the test set. Singular values were parameterized as Softmax if not otherwise stated. We kept our experimental settings the same as LoRA. Specifically, we kept hyperparameters such as sequence length, batch size, LoRA rank, LoRA alpha, and label smoothing exactly the same as LoRA. These experimental settings allow for a fair comparison with LoRA and other adaptation methods.

**Main Results.** Table 3 shows the results of GPT-2 medium/large on the E2E test set. Our method outperforms LoRA and other methods on all metrics for both GPT-2 M and GPT-2 L. The results demonstrate the effectiveness of our method in Natural Language Generation (NLG) downstream tasks and the generalization capabilities of our method across different models and task types.

## 4.4 ANALYSIS

**Scaling Up to DeBERTa-XXL.** We use DeBERTa-v2-xxlarge(1.5B) to evaluate the scaling-up performance of our method. The study was performed on three datasets of the GLUE benchmark due to the constraint of computational resources for keeping the same experimental settings with LoRA. Results in Table 4 show that BiLoRA achieves better or on par performance compared with LoRA and full fine-tuning (FT), indicating that BiLoRA yields better generalization when applied to fine-tuning models with a very large number of parameters.

Table 4: Experiment results for scaling up to DeBERTa-XXL ($D_{v2}$). In BiLoRA, the values of hyperparameters including LoRA rank, LoRA alpha, and max length are the same as those in LoRA. * indicates numbers published in prior works.

| Method | params | MNLI | MRPC | CoLA | Avg. |
|---|---|---|---|---|---|
| $D_{v2}(FT)^*$ | 1500.0M | 91.8 | 92.0 | 72.0 | 85.3 |
| $D_{v2}(LoRA)^*$ | 4.7M | $\mathbf{91.9}_{\pm.2}$ | $92.6_{\pm.6}$ | $72.4_{\pm1.1}$ | 85.6 |
| $D_{v2}(BiLoRA)$ | 4.7M | $\mathbf{91.9}_{\pm.3}$ | $\mathbf{92.7}_{\pm.4}$ | $\mathbf{73.0}_{\pm.4}$ | **85.9** |

**Ablation Studies on Pseudo Singular Values.** In Section 3.1, we introduced three ways to parameterize the pseudo singular values: Real Value, Softmax, and Approximately Binary. We conduct experiments separately using these three parameterization methods while keeping other experimental settings the same. We test RoBERTa's performance on the GLUE dataset. Results in Table 5 show that the Softmax parameterization exhibits the best performance, with Approximately Binary coming in a close second. Softmax and Approximately Binary outperform Real Value because they yield positive values which meet the constraint that singular values need to be non-negative while Real Value does not. Approximately Binary performs slightly worse than Softmax since it imposes a stronger constraint that the values need to be close to zero or one. Such a constraint limits the expressivity of the parameterization. Another observation is that under all the three parameterization methods, BiLoRA outperforms LoRA, demonstrating that BiLoRA is robust against different ways of representing the pseudo singular values and thus does not require extensive tuning for selecting the best parameterization.

Table 5: Experiment results on three different parameterizations of pseudo singular values: Real Value, Softmax, and Approximately Binary.

| Method | MNLI | SST-2 | MRPC | CoLA | QNLI | QQP | RTE | STS-B | Avg. |
|---|---|---|---|---|---|---|---|---|---|
| $R_b(LoRA)$ | 87.5 | **95.1** | 89.7 | 63.4 | **93.3** | 90.8 | 86.6 | 91.5 | 87.2 |
| $R_b(Real Value)$ | 87.5 | 94.6 | 91.7 | 63.6 | 93.0 | 90.8 | 86.6 | 91.3 | 87.4 |
| $R_b(Softmax)$ | **87.9** | **95.1** | **91.7** | **64.8** | **93.3** | **91.4** | **87.2** | **91.7** | **87.9** |
| $R_b(Binary)$ | 87.6 | 94.8 | 91.4 | 64.4 | 93.0 | 91.2 | 86.6 | 91.5 | 87.6 |

**Ablation Study on Orthogonality-Promoting Regularization.** We investigated how the tradeoff parameter $\gamma_1$ associated with the orthogonality-promoting regularizer $R_1$ in Eq.(1) affects the performance of our method. The study was performed on RoBERTa-base. Results in Table 6 show that our method is robust against different values of $\gamma_1$, which implies that using our method does not need to extensively tune this hyperparameter.

Table 6: Experiment results of RoBERTa$_{base}$ (R$_b$) on GLUE, under different values of $\gamma_1$.

| Method | MNLI | SST-2 | MRPC | CoLA | QNLI | QQP | RTE | STS-B | Avg. |
|---|---|---|---|---|---|---|---|---|---|
| R$_b(\gamma_1 = 0.0)$ | 87.8 | 95.0 | 91.7 | **64.8** | 93.1 | **91.5** | 87.2 | **91.7** | **87.9** |
| R$_b(\gamma_1 = 0.1)$ | **87.9** | **95.1** | 91.7 | **64.8** | **93.3** | 91.4 | 87.2 | **91.7** | **87.9** |
| R$_b(\gamma_1 = 0.2)$ | 87.8 | 95.0 | **91.9** | 64.4 | 93.1 | 91.2 | 86.9 | 91.5 | 87.7 |
| R$_b(\gamma_1 = 0.3)$ | 87.2 | 94.6 | 91.4 | 63.6 | 92.8 | 90.9 | **87.4** | 91.2 | 87.4 |

**Computation Costs.** Table 7 shows the training time of LoRA and our method. The total training time of our method on the eight datasets is lower than that of LoRA. This arises from the fact that BiLoRA converges with much fewer training epochs than LoRA. In the Softmax parameterization of pseudo singular values, each value is initialized with a mean equal to $1/r$, larger than that in Real-Value, which increases the overall magnitude of $\Delta W$ and allows a larger learning rate for the training process. The bi-level optimization framework effectively accommodates this larger learning rate by iteratively optimizing between the two levels without affecting the training stability. With such a large learning rate, even though bi-level optimization takes longer time for each training step, it takes much fewer training steps for training low-rank adapters compared to LoRA, thus reducing the total training time.

Table 7: Training time (minutes) of LoRA and BiLoRA on RoBERTa$_{base/large}$ (R$_{b/l}$) and the GLUE benchmark.

| Method | MNLI | SST-2 | MRPC | CoLA | QNLI | QQP | RTE | STS-B | Total. |
|---|---|---|---|---|---|---|---|---|---|
| R$_b$(LoRA) | 3190.7 | 1096.2 | **30.2** | **193.0** | 709.8 | 2464.3 | **55.5** | **62.4** | 7802.1 |
| R$_b$(BiLoRA) | **1407.1** | **260.1** | 240.3 | 260.3 | **375.2** | **1732.6** | 97.5 | 158.3 | **4531.4** |
| R$_l$(LoRA) | 789.7 | **133.9** | **14.7** | **34.1** | 209.1 | 1446.7 | 10.0 | **23.1** | 2661.3 |
| R$_l$(BiLoRA) | **707.5** | 160.8 | 19.2 | 62.5 | **200.4** | **1166.7** | **4.4** | 43.3 | **2363.8** |

The results in Table 1 and 4 jointly demonstrate that BiLoRA enhances training performance while reducing the overall training time. These results substantiate the effectiveness of our method.

## 5 CONCLUSION AND FUTURE WORK

We propose BiLoRA, a novel and general bi-level optimization framework for further enhancing the performance of low-rank adapters through addressing the overfitting issue in LoRA and its variants. By utilizing the SVD parameterization form of low-rank incremental matrices, our method separately trains pseudo singular vectors and singular values on different datasets in two different optimization levels. Such a method effectively alleviates overfitting and enhances the performance of low-rank incremental matrices while reducing the total training time. Results of extensive experiments on various NLU and NLG tasks and different large pre-trained models show that our method achieves notable performance improvements over existing adaptation methods.

Our method opens up several potential directions for future research: 1) The parameterization form of pseudo singular values can be further developed to support automated rank selection. 2) Our bi-level optimization framework enhances the generalization capability of fine-tuned models, which encourages further in-depth theoretical analysis in this regard.

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

# A DATASETS AND MODELS

## A.1 NATURAL LANGUAGE UNDERSTANDING

**GLUE Benchmark** comprises a diverse array of natural language understanding tasks widely employed for evaluation. It encompasses two single-sentence classification tasks, three tasks assessing similarity and paraphrasing, and four tasks focusing on natural language inference. Specifically, it includes MNLI (MultiNLI, Williams et al. (2017)), SST-2 (Stanford Sentiment Treebank, Socher et al. (2013)), MRPC (Microsoft Research Paraphrase Corpus, Dolan & Brockett (2005)), CoLA (Corpus of Linguistic Acceptability, Warstadt et al. (2019)), QNLI (Question NLI, Rajpurkar et al. (2018)), QQP (Quora Question Pairs), RTE (Recognizing Textual Entailment), and STS-B (Semantic Textual Similarity Benchmark, Cer et al. (2017)). We summarized the statistical data for all datasets within the GLUE Benchmark in the table below:

Table 8: The statistical data for all datasets within the GLUE Benchmark

| Dataset | Metrics | Train | Dev | Test | Label | Task |
|---------|---------|-------|-----|------|-------|------|
| MNLI | Accuracy | 393k | 20k | 20k | 3 | NLI |
| SST-2 | Accuracy | 67k | 872 | 1.8k | 2 | Sentiment |
| MRPC | Accuracy | 3.7k | 408 | 1.7k | 2 | Paraphrase |
| CoLA | Matthews corr | 8.5k | 1k | 1k | 2 | Acceptability |
| QNLI | Accuracy | 108k | 5.7k | 5.7k | 2 | QA/NLI |
| QQP | Accuracy | 364k | 40k | 391k | 2 | Paraphrase |
| RTE | Accuracy | 2.5k | 276 | 3k | 2 | NLI |
| STSB | Pearson corr | 7.0k | 1.5k | 1.4k | 1 | Similarity |

## A.2 NATURAL LANGUAGE GENERATION

**E2E NLG Challenge** (Novikova et al., 2017) is now commonly used for data-to-text evaluation. It was first introduced as a dataset for training end-to-end, data-driven natural language generation systems. Multiple references can be associated with each source table used as input. Each sample input $(x, y)$ is composed of a series of slot-value pairs, accompanied by an associated natural language reference text. The E2E dataset consists of approximately 42,000 training examples, 4,600 validation examples, and 4,600 test examples from the restaurant domain.

## A.3 MODELS

**RoBERTa** (Liu et al., 2019) builds upon the foundational principles and training strategies of BERT (Devlin et al., 2018), offering novel alternatives that enhance downstream task performance. RoBERTa refines and optimizes the pre-training methodology initially proposed in BERT, resulting in notable improvements in task performance while maintaining a comparable number of trainable parameters. We use RoBERTa-base and RoBERTa-large for a convenient and fair comparison with LoRA (Hu et al., 2021).

**DeBERTa** (He et al., 2020) represents an advanced iteration of BERT, having undergone extensive training at a larger scale. DeBERTa demonstrates strong competitiveness when evaluated on the GLUE benchmark. For our experiments, we use DeBERTa-v2-xxlarge which has 1.5 billions of parameters to evaluate the scaling-up capability of BiLoRA and also for a convenient comparison with LoRA. We use DeBERTa-v3-base which has 183 millions parameters for fair comparison with AdaLoRA (Zhang et al., 2023).

**GPT-2** (Radford et al., 2019) developed by OpenAI, was once a state-of-the-art language model renowned for its remarkable text generation capabilities. It is a scaled-up version of its predecessor, GPT-1, and is trained on an extensive corpus of text data. GPT-2 has been widely recognized for its proficiency in generating coherent and contextually relevant text across various natural language

understanding and generation tasks, showcasing its versatility and potential in the field of natural language processing.

# B EXPERIMENTAL SETTINGS

## B.1 ROBERTA

We summarized the experimental settings for the experiments of RoBERTa-base and RoBERTa-large in Table9. In fact, we only introduced an additional level of learning rate compared to LoRA. For hyperparameters such as max seq length, LoRA $\alpha$, we kept them the same as LoRA. We chose learning rates from the magnitude of 1e-5 for almost all of our experiments. The hyperparameter tuning for our method is quite simple, convenient and straightforward.

Table 9: The hyperparameters we used for RoBERTa on the GLUE benchmark. $*$ indicates model already adapted to MNLI when adapting to MRPC, RTE, and STS-B, while $\dagger$ indicates model started as pre-trained when adapting to all datasets.

| Method | Settings | MNLI | SST-2 | MRPC | CoLA | QNLI | QQP | RTE | STS-B |
|---|---|---|---|---|---|---|---|---|---|
| | Optimizer | \multicolumn{8}{c}{AdamW} | | | | | | | |
| | Warmup Ratio | 0.06 | | | | | | | |
| | Scheduler | Linear | | | | | | | |
| | LoRA rank | $rank_q = rank_v = 8$ | | | | | | | |
| RoBERTa-base$^*$ | Total batch size | 64 | | | | | | | |
| | Global steps | 10k | 3k | 2k | 3k | 3k | 15k | 1.5k | 5k |
| | Lower learning rate | 2e-5 | 3e-5 | 2e-6 | 2e-5 | 3e-5 | 3e-5 | 4e-6 | 4e-6 |
| | Upper learning rate | 3e-5 | 4e-5 | 8e-6 | 4e-5 | 4e-5 | 4e-5 | 2e-6 | 2e-6 |
| | Lower weight decay | 0.12 | 0.12 | 0.12 | 0.12 | 0.12 | 0.12 | 0.12 | 0.1 |
| | Upper weight decay | 0.1 | 0.1 | 0.1 | 0.1 | 0.1 | 0.1 | 0.1 | 0.1 |
| | Max Seq Length | 512 | | | | | | | |
| RoBERTa-large$^*$ | Total batch size | 32 | | | | | | | |
| | Global steps | 15k | 4k | 1k | 3k | 3k | 20k | 0.12k | 2k |
| | Lower learning rate | 1.5e-5 | 1.5e-5 | 4e-6 | 1e-5 | 1e-5 | 1e-5 | 4e-6 | 1e-5 |
| | Upper learning rate | 2e-5 | 2e-5 | 6e-6 | 5e-5 | 3e-5 | 2e-5 | 4e-6 | 5e-6 |
| | Lower weight decay | 0.12 | 0.12 | 0.12 | 0.12 | 0.12 | 0.12 | 0.1 | 0.1 |
| | Upper weight decay | 0.1 | 0.1 | 0.1 | 0.1 | 0.1 | 0.1 | 0.1 | 0.1 |
| | Max Seq Length | 128 | | | | | | | |
| RoBERTa-large$^\dagger$ | Total batch size | 32 | | | | | | | |
| | Global steps | 15k | 4k | 1k | 3k | 3k | 20k | 2k | 2k |
| | Lower learning rate | 1.5e-5 | 1.5e-5 | 2e-5 | 1e-5 | 1e-5 | 1e-5 | 1e-5 | 8e-6 |
| | Upper learning rate | 2e-5 | 2e-5 | 1e-4 | 5e-5 | 3e-5 | 2e-5 | 2e-5 | 4e-6 |
| | Lower weight decay | 0.12 | 0.12 | 0.12 | 0.12 | 0.12 | 0.12 | 0.12 | 0.1 |
| | Upper weight decay | 0.1 | 0.1 | 0.1 | 0.1 | 0.1 | 0.1 | 0.1 | 0.1 |
| | Max Seq Length | 128 | | | | | | | |

## B.2 DEBERTA

We summarized the experimental settings used in the experiments for DeBERTa-v2-xxlarge and DeBERTa-v3-base in Table10. In fact, we only introduced an additional level of learning rate compared to LoRA. For hyperparameters such as max seq length, LoRA $\alpha$, we kept them the same as LoRA and AdaLoRA. We chose learning rates from the magnitude of 1e-5 for almost all of our experiments. The hyperparameter tuning for our method is quite simple, convenient and straightforward. Due to our limited computational resources, we were unable to maintain the same experimental settings as LoRA on many datasets, making a fair comparison impossible. Therefore, for DoBERTa-v2-xxlarge, we only conducted experiments on the MNLI, CoLA, and MRPC datasets.

Table 10: The hyperparameters we used for DeBERTa-v2-xxlarge and DeBERTa-v3-base on the GLUE benchmark.

| Method | Settings | MNLI | SST-2 | MRPC | CoLA | QNLI | QQP | RTE | STS-B |
|---|---|---|---|---|---|---|---|---|---|
| | Optimizer | | | | AdamW | | | | |
| | Scheduler | | | | Linear | | | | |
| | LoRA rank | | | | $rank_q = rank_v = 8$ | | | | |
| DeBERTa-v2-XXL | Total batch size | 64 | | 32 | 32 | | | | |
| | Global steps | 20k | | 1k | 3k | | | | |
| | Inner learning rate | 0.5e-5 | | 2e-6 | 1e-5 | | | | |
| | Outer learning rate | 1e-5 | | 2e-6 | 1e-5 | | | | |
| | LoRA $\alpha$ | 16 | | 16 | 16 | | | | |
| | Max Seq Length | 128 | | 128 | 64 | | | | |
| DeBERTa-v3-base | Total batch size | | | | 32 | | | | |
| | Global steps | 15k | 3k | 1k | 1k | 2k | 20k | 0.5k | 1k |
| | Lower learning rate | 1e-5 | 1.5e-5 | 2e-6 | 1e-5 | 1e-5 | 1e-5 | 4e-6 | 4e-6 |
| | Upper learning rate | 2e-5 | 2.5e-5 | 4e-6 | 2e-4 | 2e-5 | 2e-5 | 4e-6 | 4e-6 |
| | Lower weight decay | 0.12 | 0.12 | 0.15 | 0.12 | 0.12 | 0.12 | 0.12 | 0.12 |
| | Upper weight decay | 0.1 | 0.1 | 0.1 | 0.1 | 0.1 | 0.1 | 0.1 | 0.1 |
| | LoRA $\alpha$ | | | | 16 | | | | |
| | Max Seq Length | 256 | 128 | 320 | 64 | 512 | 320 | 320 | 128 |

### B.3 GPT-2

We summarized the experimental settings for the experiments of GPT-2 M and L in Table11. We kept hyperparameters almost the same as LoRA for a fair comparison.

Table 11: The hyperparameters we used for GPT-2 on the E2E NLG benchmark.

| Settings | Training |
|---|---|
| Optimizer | AdamW |
| Warmup Steps | 500 |
| Scheduler | Linear |
| LoRA rank | $rank_q = rank_v = 4$ |
| LoRA $\alpha$ | 32 |
| Label Smooth | 0.1 |
| Weight Decay | 0.01 |
| Batch Size | 8 |

| Settings | Inference |
|---|---|
| Beam Size | 10 |
| Length Penalty | 0.9 |
| no repeat ngram size | 4 |

## C  MOTIVATION

Figure 2.

## D  PRUNING RATES AND WEIGHT DECAY IN ADALORA

Figure 3 and Figure 4.

## E  THE DISTRIBUTION OF THE SINGULAR VALUES

Figure 5.

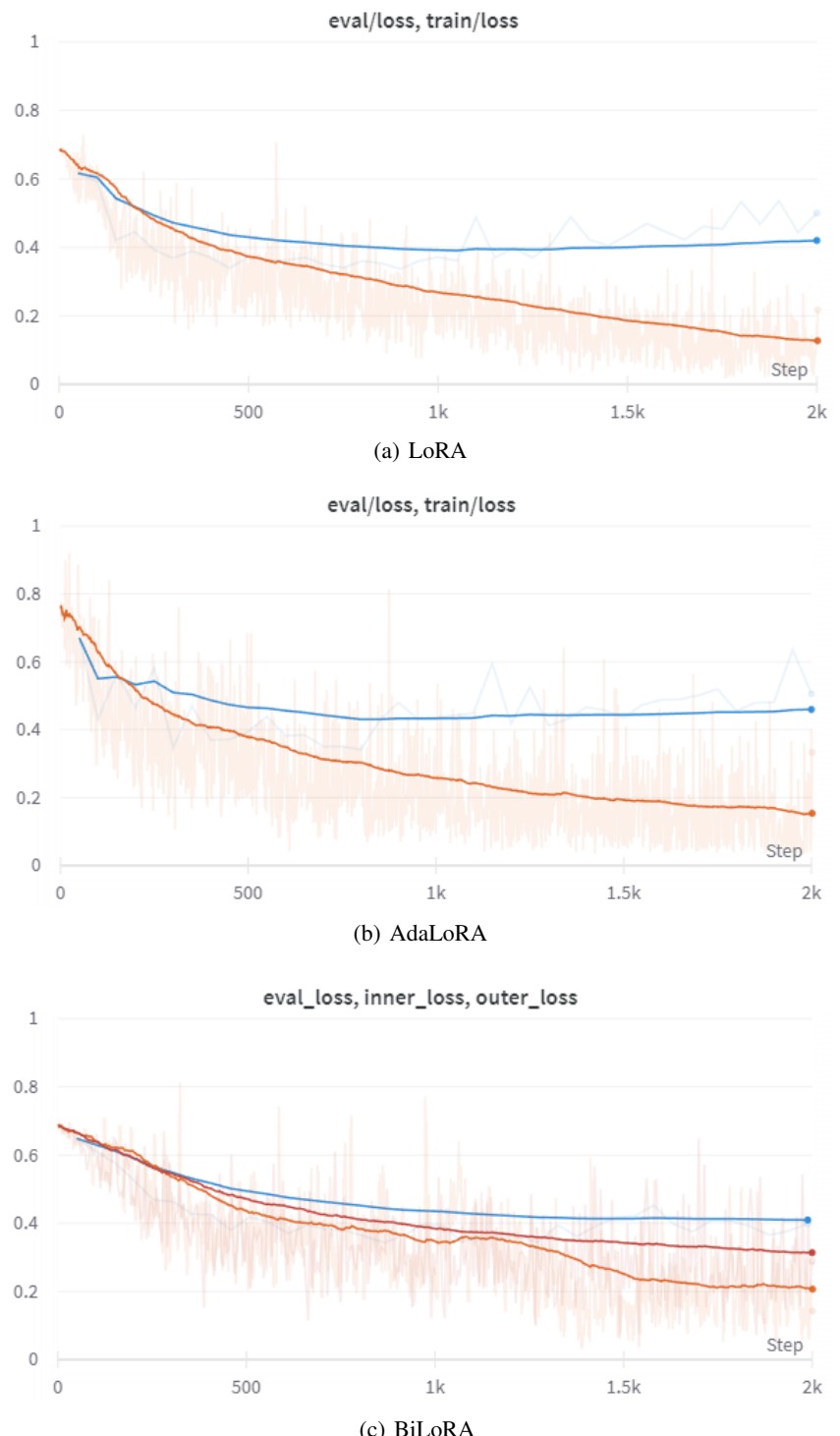

(a) LoRA

(b) AdaLoRA

(c) BiLoRA

Figure 2: Training/Evaluation Loss Curves for illustrating of the overfitting limitations of existing methods. Blue curves represent evaluation losses. In LoRA and AdaLoRA, yellow curves represent training losses while in BiLoRA yellow/red curves separately represent inner/outer losses.

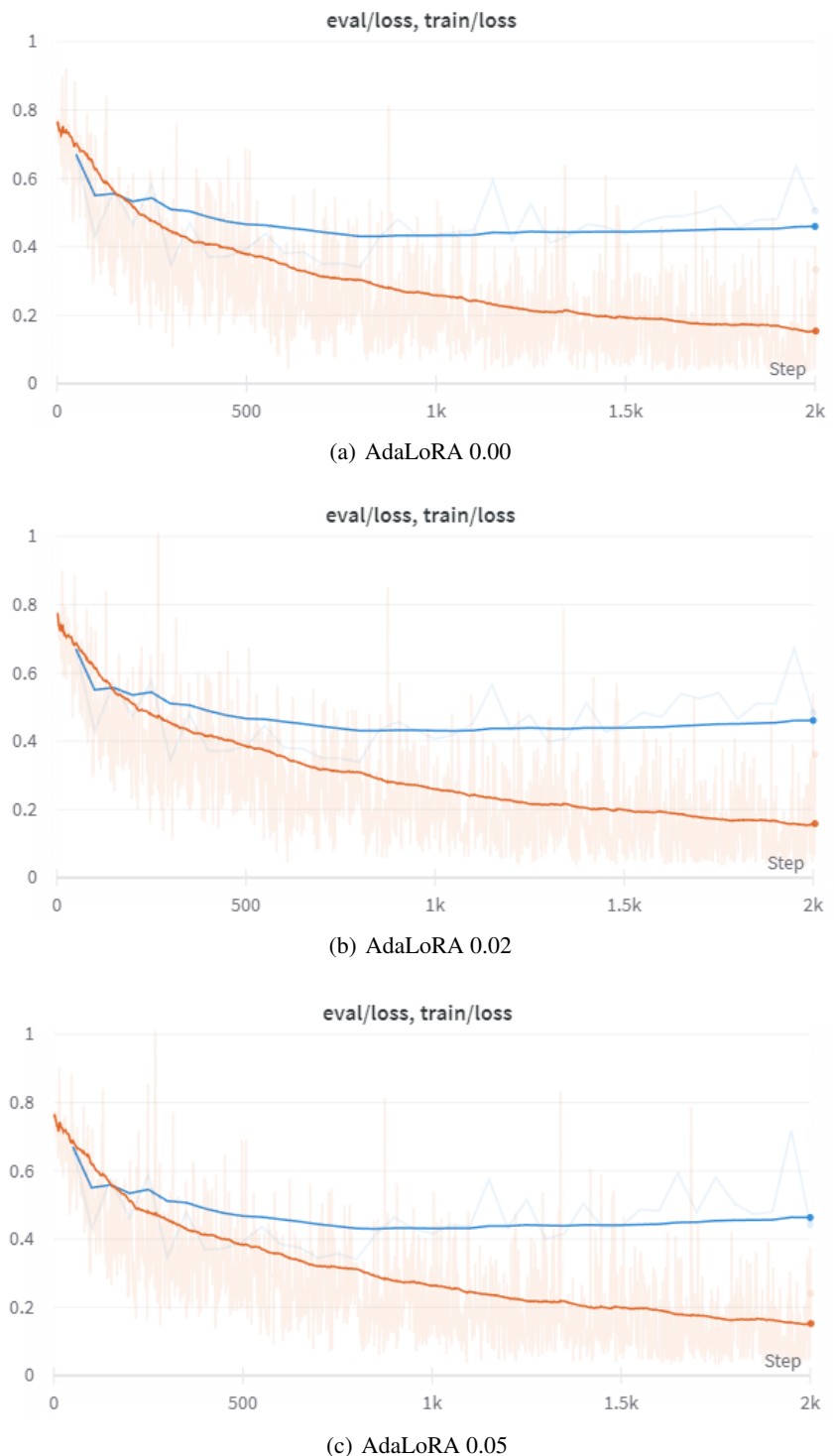

Figure 3: Training/Evaluation Loss Curves for illustrating of the influence of different weight decays in AdaLoRA. Blue curves represent evaluation losses and yellow curves represent training losses.

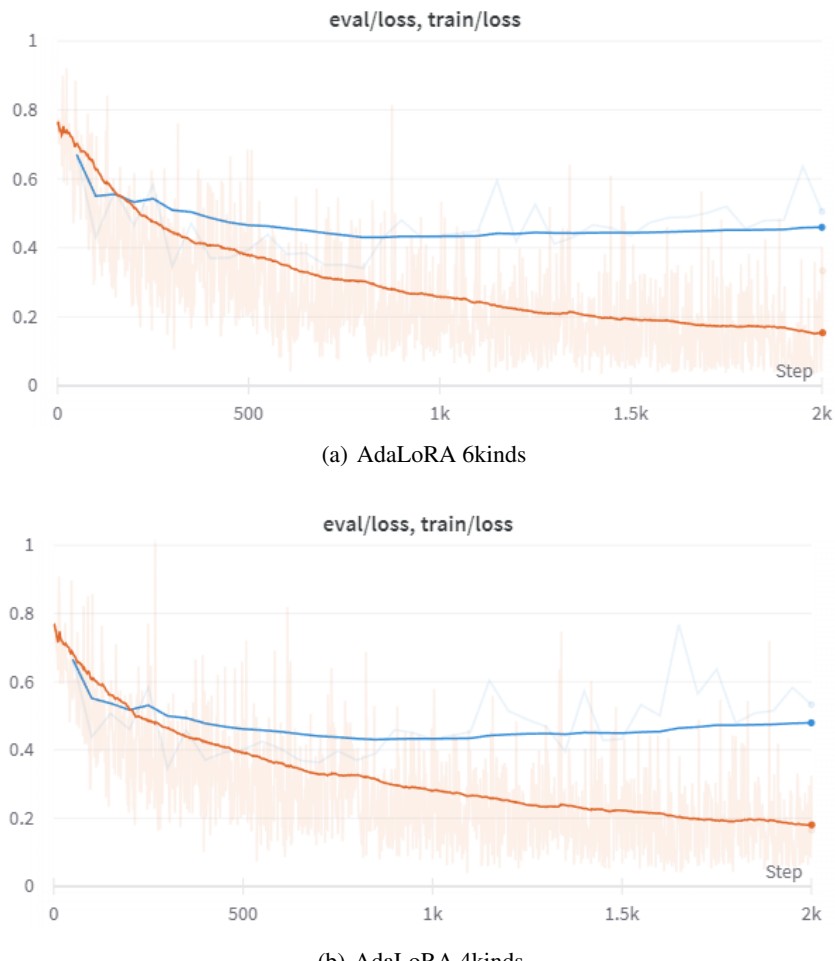

(a) AdaLoRA 6kinds

(b) AdaLoRA 4kinds

Figure 4: Training/Evaluation Loss Curves for illustrating of the influence of different pruning rates in AdaLoRA. Blue curves represent evaluation losses and yellow curves represent training losses.

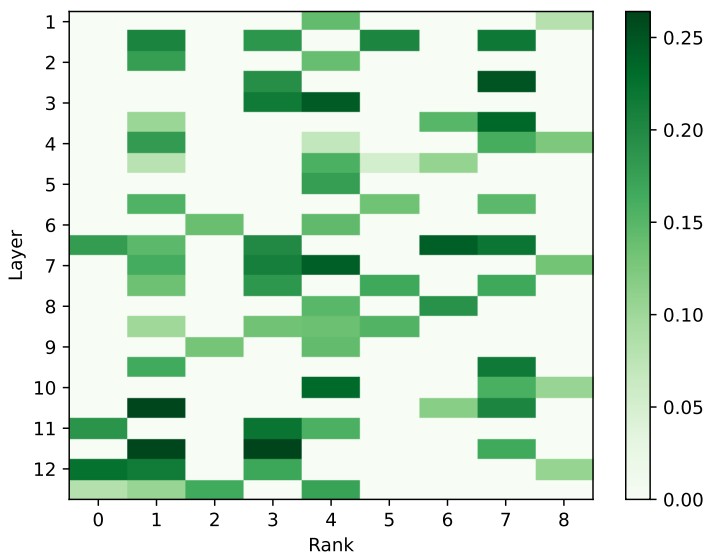

(a) AdaLoRA Singular Value Distribution

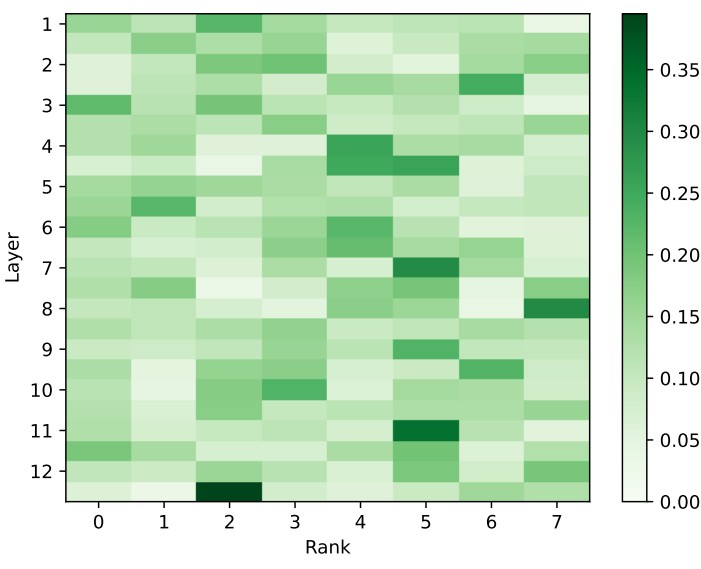

(b) BiLoRA Singular Value Distribution

Figure 5: Singular Value Distribution of BiLoRA and AdaLoRA.

# F    THE ORTHOGONALITY OF OPTIMAL SOLUTIONS

Figure 6 and Figure 7.

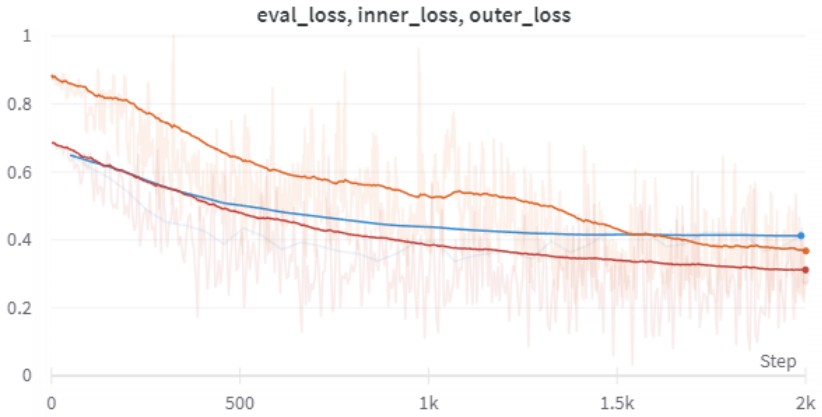

Figure 7: Loss Curve with $\lambda = 0.1$

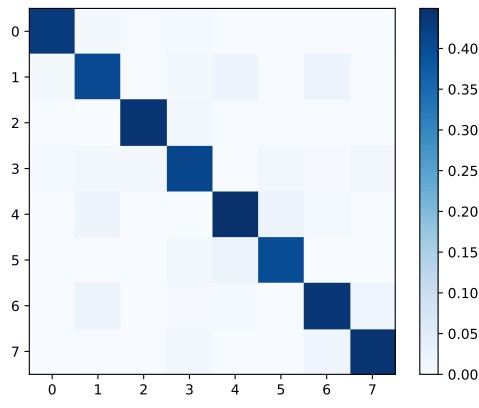

(a) BiLoRA Without Regularization

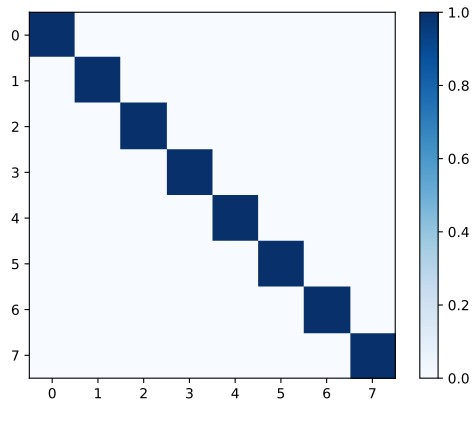

(b) BiLoRA With $\lambda = 0.1$

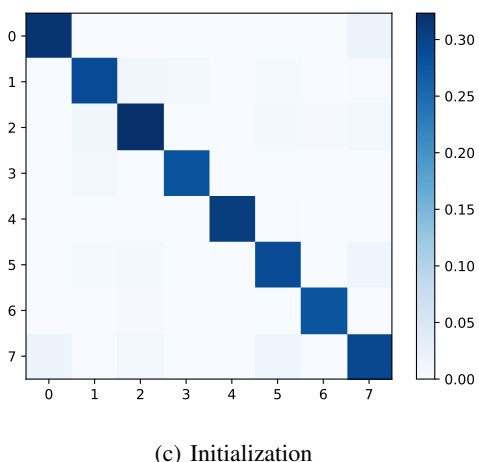

(c) Initialization

Figure 6: The orthogonality of optimal solutions.

