# OpenReview forum: "BiLoRA: A Bi-level Optimization Framework for Low-rank Adapters"
_ICLR.cc/2024/Conference — Submitted to ICLR 2024_

### Official Review · Reviewer_MGLE · 2023-10-26

**Soundness:** 2 fair
**Presentation:** 3 good
**Contribution:** 2 fair
**Rating:** 5
**Confidence:** 4

**Summary:**

This paper considers parameter efficient fine-tuning built on low rank adaptation (LoRA) and its adaptive extension AdaLoRA. The main limitations addressed regarding LoRA and AdaLoRA are that (1) LoRA uses a constant rank for all adaptors, whereas pre-trained weight matrices may have varying levels of importance for a downstream task and (2) AdaLoRA searches for pseudo singular values/vectors on the entire training set. It is suggested that (2) results in overfitting, and the main idea behind the proposed BiLoRA is to train singular values and vectors on a different partition of the training set.

**Strengths:**

* This paper conducts several ablation studies for various constraints that can be imposed on the singular values/vectors.
* Proposed method consumes much less training time as seen in Table 7.
* Various (large) architectures are tested on natural language understanding and language generation to demonstrate the effectiveness of the proposed method: RoBERTa base/large; GPT-2 medium/large; DeBERTa-XXL (1.5B params).
* Empirical conclusions are sound, in particular how unconstrained pseudo-singular values that allow for negative singular values result in worse performance than the variant that uses softmax which results in non-negative singular values.

**Weaknesses:**

* My main concern is that there is no evidence to support the claim the basis of this paper, that AdaLoRA overfits because it trains both singular values and vectors on the same full training set. While the authors claim this to motivate the need for bi-level optimization, it is unclear that the basis of their work is an actual limitation of existing methods. Without any evidence on the alleged limitations of AdaLoRA, it is unclear what advantages the bi-level optimization approach brings.
* Although the main methodological difference with AdaLoRA is the bi-level optimization approah, the paper doesn't really experiment with the associated hyper-parameters. How is the number of iterations for singular value/vector updates found? Does this result in large performance/training-time differences? If the results are robust to the choice of $T_1$ and $T_2$, why not alternate between the two, i.e. $T_1 = T_2 = 1$? How does the dataset partition (choice of data on which singular values and singular vectors are trained on) affect the performance and training times?

**Questions:**

* Ablation study on orthogonality regularization for the singular vectors illustrate that the performance is largely unaffected by the coefficient of this regularization, i.e. penalty for singular vectors violating orthonormality has minimal effect. Is this because the resulting pseudo-singular vectors turn out to be nearly orthogonal? It would be nice to see if the optimal solutions found are nearly orthogonal by plotting their angles & magnitudes.

---

> ### Author Response · Authors · 2023-11-16
> **Thank you for your constructive feedback.**
>
> We appreciate your constructive feedback very much. We provide all the figures from Figure 2-7 in the Supplementary Section C-F of our paper. All the indexes of figures are referred to those of the updated paper. We sincerely refer you to the figures in our paper. We provide our response to your questions as follows.
>
> >Weakness 1: The basis of this paper.
>
> The basis of BiLoRA can be concluded in 4 aspects.
> * Limitations of existing fine-tuning methods.
> * Inspirations and advantages of bi-level optimization.
> * Accelerating the fine-tuning process.
> * Opportunities for deeper understanding of PEFT process.
>
> First, existing fine-tuning methods can suffer from overfitting. The large number of parameters in pre-trained models may make the full fine-tuning process more prone to overfitting[1]. Existing PEFT methods do not well solve this problem. The generalization capability of a fine-tuned model on the evaluation dataset can reflect the range of overfitting. We plot the training loss and evaluation loss of LoRA, AdaLoRA and BiLoRA to illustrate the generalization capability. We conduct experiments on RoBERTa-base on CoLA dataset and clip the first 2k steps. Figure 2(a) and Figure 2(b) show that for both LoRA and AdaLoRA, the evaluation loss starts to increase at or before 1k step while the training loss still decreases. The gap between the 2 losses is getting huge. This demonstrates the limitations of existing methods that they still suffer from overfitting issues. Figure 2(c) shows that both inner training loss and outer training loss are close to the evaluation loss. The evaluation loss is still decreasing after 2k steps and is obviously lower than that of LoRA and AdaLoRA. These results convincingly demonstrate the limitations of existing methods and the effectiveness of BiLoRA in addressing overfitting.
>
> Second, we gain inspiration from DARTS (Differentiable Architecture Search) which utilizes a bi-level optimization for searching network architectures. DARTS uses training dataset and validation dataset separately for training operations and their corresponding proportions in 2 levels. DARTS conducts experiments on comparison between bi-level optimization and the normal training method(single-level). The latter performs much worse and they concludes that the latter “would cause α (analogous to hyperparameters) to overfit the training data, leading to poor generalization”[2]. We gain this inspiration of preventing overfitting and innovatively explore the possibilities of applying bi-level optimization to PEFT domain for addressing the existing limitations of overfitting.
>
> Third, one important goal of PEFT is to accelerate the fine-tuning process. The bi-level optimization framework effectively accommodates larger learning rates by iteratively optimizing between the two levels without affecting the training stability. Through BiLoRA, we significantly reduce the training steps and training time needed for convergence.
>
> Fourth, we expect BiLoRA to encourage deeper understanding of low-rank fine-tuning processes and more design potentials. We offer more flexible and direct interaction with singular values and singular matrices at separate levels and design different parameterization of singular values. More efficient methods and deeper insights can be inspired by this bi-level optimization framework.
>
> >Weakness 2: The influence of the dataset partition.
>
> **Data Partition:**
> The Dataset Partition, together with learning rate can help keep the balance of inner/outer optimization, which can contribute greatly to preventing the model from overfitting. Lower level has more trainable parameters, so it’s natural to use more data for training singular vectors, while using the left for training singular values. We further experiment on DeBERTa-v3-base on CoLA and SST2 datasets to show the influence of different dataset partitions. We change the inner level dataset partition from 0.6 to 1.0 with 0.1 interval.
> | Data Partition | CoLA  | SST2  |
> |----------------|-------|-------|
> | 0.6            | 68.01 | 94.84 |
> | 0.7            | 70.16 | 95.87 |
> | 0.8            | **70.52** | 96.02 |
> | 0.9            | 70.39 | **96.44** |
> | 1.0            | 67.94 | 94.51 |
>
> Results show that too small partitions(<=0.6) or too large partitions(1.0, only train singular matrix) can harm the overall performance.When the inner partition is too small, singular vectors are not well trained and when the inner partition is 1.0, singular vectors are not trained, which can cause a great performance drop. These results can also show that the two levels are both necessary in preventing overfitting and enhancing performances.
> In the paper, we don’t ever change the partition of the data and keep it 8:2.
>
> [1] Rabeeh Karimi Mahabadi, James Henderson, and Sebastian Ruder. Compacter: Efficient low-rank hypercomplex adapter layers.
>
> [2] Liu H, Simonyan K, Yang Y. Darts: Differentiable architecture search.

---

> ### Author Response · Authors · 2023-11-16
> **Response to other constructive feedback and questions.**
>
> >Weakness 2 addition: The influence of $T_1/T_2$.
>
> **Number of iterations:**
> In the paper, we didn’t change the iteration numbers throughout our experiments and kept T1=T2=1. Actually there can be a set of hyperparameters that can all achieve good results because they can all keep good balance of inner/outer level and achieve 3 perspectives. 1) performing well on inner dataset; 2) performing well on outer dataset; 3) not overfitting on either subset and generalizing well on test dataset.
> We further experiment on different iteration numbers as follows. We experiment on DeBERTa-v3-base on CoLA dataset. For larger T1 and T2, one global step contains more training steps so we reduce the total global steps accordingly. All other hyperparameters are kept the same.
>
> | T1/T2 | 1     | 3     |
> |-------|-------|-------|
> | 1     | 70.52 | 70.40 |
> | 2     | 69.96 | 70.14 |
> | 5     | 70.01 | 69.19 |
>
> The total optimization steps for inner level and outer level are close for different T1/T2. Typically, a single inner optimization step is faster than a single outer optimization step due to the calculation of hypergradients of outer level. So using a larger T1 is also an efficient choice.
>
> We don’t exactly tune the iteration numbers for 2 reasons.
>
> * T1=T2=1 is an empirical choice in existing bi-level optimization tasks. The main effect of iteration numbers can be the balance of inner and outer level. Intuitively and practically, T1=T2=1 means BiLoRA frequently optimizes between the 2 levels, preventing the model from overfitting on either subset and effectively addresses overfitting.
>
> * Simplicity. We expect BiLoRA to be powerful and effective, yet easy to use. The number of hyperparameters for BiLoRA is kept nearly the same as LoRA, which is much less than AdaLoRA.
>
> >Question 1: The orthogonality of optimal solutions.
>
> We suspect the reason for “the performance is largely unaffected by the coefficient of this regularization” can be from these 3 aspects.
> * The value distribution of $P^{T}P$ and $QQ^{T}$ in optimal solutions with regularization coefficient $\lambda=0.0$ is close to Identity Matrix. Despite magnitudes, singular vectors are largely orthogonal. When regularization coefficient $\lambda >= 0.1$, singular matrices are almost orthogonal.
> * This “natural orthogonality" without regularization can be kind of due to the Normal Initialization we use for initializing all the singular vector matrices.
> * Optimal solutions with different regularization coefficients including $\lambda=0.0$ can all contribute greatly to prevent overfitting.
>
> First, we sample one of the singular vector matrices from the optimal solutions with reg coefficient=0.0 and 0.1. Since we use rank=8, the dimension of the resulting matrix is 8×8. We plot $P^{T}P$ in both AdaLoRA and BiLoRA. We sincerely refer you to the plots in Figure 6 (a)(b). The plots show that the value distribution of $P^{T}P$ from optimal solutions with reg coefficient=0.0  is very close to the Identity Matrix. When the reg coefficient > 0.0, the pseudo-singular vectors turn out to be almost orthogonal. These results can explain the robustness of BiLoRA across different regularization coefficients.
>
> Secondly, this “natural orthogonality" can be kind of due to the Normal Initialization. We further plot the sampled matrix just after initialization. We sincerely refer you to the plots in Figure 6 (c). We use “nn.init.normal_()” for initializing all the singular vector matrices.
>
> Third, optimal solutions with different reg coefficients can all contribute greatly to prevent overfitting. We plot the loss curves with reg coefficient=0.0 and 0.1. We sincerely refer you to the curve in Figure 7 and Figure 2.
>
> Results demonstrate that with different regularization coefficients, BiLoRA can effectively prevent overfitting.

---

> > ### Comment · Reviewer_MGLE · 2023-11-22
> >
> > I appreciate the detailed response. I agree with Reviewer YcPD on the motivation for bi-level optimization. The authors confirmed the empirical benefits, but it's still not clear why bi-level optimization fixes this overfitting behavior. However, I think the responses addressed many concerns I had so I raise my score to 5. Reason to not go further up is that even with the response, I don't see a compelling explanation for *why* any of the techniques used works out well in experiments.

---

> ### Author Response · Authors · 2023-11-22
>
> Thank you for all the constructive feedback. We really appreciate your insightful suggestions.
>
> We highly value your concerns regarding the motivation and the reason why techniques work well, and we genuinely provide the following response in the hope of alleviating your concerns.
>
> Extensive experiment results demonstrate the better generalization capability of BiLoRA and loss curves convincingly illustrate that BiLoRA largely addresses the overfitting issue.
> There may be 3 main useful techniques for the improvement. 1) Bi-level optimization framework. 2) Parameterization of Singular Values. 3) Orthogonality Initialization and Regularization.
>
> **Bi-level Optimization Framework.**
>
> As far as we are concerned, the reason for utilizing bi-level optimization for low-rank adaptation and the powerful effect is mainly from 3 aspects:
> * The overfitting issue has long been a hindrance to the further improvement of existing fine-tuning methods.
> * Convincing results, conclusions and inspirations from DARTS. DARTS conducts experiments on comparison between bi-level optimization and the normal training method(single-level). Results demonstrate that the latter performs much worse and DARTS concludes that the latter “would cause α (analogous to hyperparameters) to overfit the training data, leading to poor generalization”. Bi-level optimization has been shown to improve the generalization capability.
> * The nature of bi-level optimization. Iteratively optimizing between the 2 levels can intuitively prevent overfitting on either of the 2 subsets.
>
> **Parameterization of Singular Values.**
>
> The parameterization of singular values can help improve the performance because:
> *  Softmax and Approximately Binary Parameterization outperform Real Value (the parameterization in LoRA and AdaLoRA) because they yield positive values which meet the constraint that singular values need to be non-negative while Real Value does not.
>
> **Orthogonality Initialization and Regularization.**
>
> The Orthogonality Initialization and Regularization can help improve the performance due to:
> * With $P$ and $Q$ to be almost orthogonal, $\Lambda$ can be exactly the singular value matrix of the low-rank adapters according to SVD. Thus the outer level is directly optimizing the singular values which can be viewed as importance of diifferent ranks.
> * The Orthogonality Regularization can contribute to preventing overfitting since it helps optimizing the training loss while still keeping the evaluation loss decreasing rather than overfitting. This has been convincingly illustrated by the loss curves.
>
>
> BiLoRA innovatively introduces bi-level optimization to the Parameter-Efficient-Fine-Tuning domain. It explores and demonstrates the huge potentials of utilizing bi-level optimization for improving the generalization capability of low-rank adapters and speeding up the fine-tuning process. We trust BiLoRA will help enhance the efficiency and performance of many models on many downstream tasks.
>
> Thank you again for engaging with us and your valuable feedback.

---

### Official Review · Reviewer_3MBr · 2023-10-29

**Soundness:** 3 good
**Presentation:** 3 good
**Contribution:** 3 good
**Rating:** 6
**Confidence:** 3

**Summary:**

The paper proposes BiLoRA that leverages bi-level optimization on different subsets of the data to resolve overfitting. The proposed method is efficient and produces better finetuning results across NLU and NLG tasks. Overall, the paper makes a valid contribution, and there are only some minor concerns.

**Strengths:**

1. Writing is pretty clear. I am able to easily follow most part of the paper.
2. The idea to train different parameters (singular values vs vectors) at different subsets of the dataset to reduce overfitting makes sense intuitively.
3. The method is efficient to train and also produces better results for both NLU and NLG.

**Weaknesses:**

1. I fail to understand why “Learning Λ by minimizing a single dataset’s training loss can easily render these contributions and parameter amounts tailored to this dataset”. In machine learning, aren’t we always learn parameters by minimizing a single dataset’s training loss?

2. It is mentioned in many places that existing methods train "on a single training dataset". "single training dataset" can be confusing here. The proposed BiLoRA even though uses different subsets, it is also still trained on the same single dataset.

3. Is the motivation that AdaLoRA can overfit to training set? If overfitting happens, can you just prune the singular values more aggressively? What happen if you just use larger weight decay in AdaLoRA?

4. Can you compare the distribution of the singular values learned in BiLoRA vs AdaLoRA? This can be helpful to understand BiLoRA more.

**Questions:**

see weakness

---

> ### Author Response · Authors · 2023-11-16
> **Thank you for your constructive feedback.**
>
> We appreciate your constructive feedback very much. Since we are not able to submit a PDF, we provide all the figures from Figure 2-7 in the Supplementary Section C-F of our paper. All the indexes of figures are referred to those of the updated paper. We sincerely refer you to the figures in our paper. We provide our response to your questions as follows.
>
> >Weakness 1 and 3: The limitation of minimizing a single dataset’s training loss and the motivation of BiLoRA.
>
> The motivation of BiLoRA can be concluded in 4 aspects.
> * Limitations of existing fine-tuning methods.
> * Inspirations and advantages of bi-level optimization.
> * Accelerating the fine-tuning process.
> * Opportunities for deeper understanding of PEFT process.
>
> First, existing fine-tuning methods can suffer from overfitting. The large number of parameters in pre-trained models may make the full fine-tuning process more prone to overfitting[1]. Existing PEFT methods do not well solve this problem. The generalization capability of a fine-tuned model on the evaluation dataset can reflect the range of overfitting. We plot the training loss and evaluation loss of LoRA, AdaLoRA and BiLoRA to illustrate the generalization capability. We conduct experiments on RoBERTa-base on CoLA dataset and clip the first 2k steps. Figure 2(a) and Figure 2(b) show that for both LoRA and AdaLoRA, the evaluation loss starts to increase at or before 1k step while the training loss still decreases. The gap between the 2 losses is getting huge. This demonstrates the limitations of existing methods that they still suffer from overfitting issues. Figure 2(c) shows that both inner training loss and outer training loss are close to the evaluation loss. The evaluation loss is still decreasing after 2k steps and is obviously lower than that of LoRA and AdaLoRA. These results convincingly demonstrate the limitations of existing methods and the effectiveness of BiLoRA in addressing overfitting.
>
> Second, we gain inspiration from DARTS (Differentiable Architecture Search) which utilizes a bi-level optimization for searching network architectures. DARTS uses training dataset and validation dataset separately for training operations and their corresponding proportions in 2 levels. DARTS conducts experiments on comparison between bi-level optimization and the normal training method(single-level). The latter performs much worse and they concludes that the latter “would cause α (analogous to hyperparameters) to overfit the training data, leading to poor generalization”[2]. We gain this inspiration of preventing overfitting and innovatively explore the possibilities of applying bi-level optimization to PEFT domain for addressing the existing limitations of overfitting. Extensive experiments demonstrate the effectiveness of BiLoRA.
>
> Third, one important goal of PEFT is to accelerate the fine-tuning process. The bi-level optimization framework effectively accommodates larger learning rates by iteratively optimizing between the two levels without affecting the training stability. Through BiLoRA, we significantly reduce the training steps and training time needed for convergence, thus making PEFT more efficient.
>
> Fourth, we expect BiLoRA to encourage deeper understanding of low-rank fine-tuning processes and more design potentials. We offer more flexible and direct interaction with singular values and singular matrices at separate levels and design different parameterization of singular values. More efficient methods and deeper insights can be inspired by this bi-level optimization framework.
>
> >Weakness 2: The clarification of "single training dataset".
>
> When we mention that existing methods train "on a single training dataset", we mean training on a single training dataset in a single training process. Compared with this common practice in machine learning, we innovatively leverage the characteristics of bi-level optimization, optimizing low-rank adapters iteratively between upper and lower levels, to address the overfitting issues that arise in existing fine-tuning methods on a single dataset during a single training process. BiLoRA aligns with intuition: by iteratively optimizing between two levels, the model is less prone to overfitting on each of the two sub-datasets and the whole “single dataset”, resulting in enhanced generalization capability and better performance. Extensive experiments on multiple models and datasets strongly support this observation.
>
>
> [1] Rabeeh Karimi Mahabadi, James Henderson, and Sebastian Ruder. Compacter: Efficient low-rank hypercomplex adapter layers.
>
> [2] Liu H, Simonyan K, Yang Y. Darts: Differentiable architecture search.

---

> ### Author Response · Authors · 2023-11-16
> **Response to other constructive feedback and questions.**
>
> >Weakness 3 addition: The influence of pruning rates and weight decay in AdaLoRA.
>
> The hyperparameters for AdaLoRA have been well tuned by the authors of AdaLoRA, so the results are already the best that can be achieved by AdaLoRA. We conduct several experiments trying to offer some insights about the influence of pruning rates and weight decay in AdaLoRA. We experiment on different pruning rates and larger weight decay on AdaLoRA. We experiment on DeBERTa-v3-base on the COLA dataset. Results can be seen as follows.
> |  weight decay  |  0.00 (origin)  | 0.01 | 0.02 | 0.05 |
> |--------------|-----------------------|------|------|------|
> | performance  | 70.04           | 68.91| 68.42| 68.03|
>
> When applying larger weight decay in AdaLoRA, the performance drops. We show the training/evaluation loss curves of different weight decays. We sincerely refer you to the curves in Figure 3. There is no significant difference among the curves.
>
> Pruning rates can be tuned by changing the target rank count. The target rank under current experimental settings is already 1 so we reduce the layer kinds from 6 to 4. Results can be seen as follows.
>
> | pruning rates | 6 kinds | 4 kinds |
> |---------------|---------|---------|
> | performance   | 70.04   | 67.97   |
>
> We also plot the training/evaluation loss curves of different pruning rates. We sincerely refer you to the curves in Figure 4.
>
> Both pruning and weight decay cannot largely solve the overfitting problem. So we still consider this overfitting phenomenon to be an important issue for existing methods.
>
> >Weakness 4: Comparison between the distribution of the singular values learned in BiLoRA and AdaLoRA.
>
> We plot the distribution of singular values learned both in BiLoRA and AdaLoRA. We use the singular values of RoBERTa-base on the CoLA dataset. AdaLoRA implements low-rank adapters to all linear layers including query, key, value, intermediate, layer.output and attention.output while LoRA and BiLoRA only implements low-rank adapters to query and key. We use abs for all the singular values of AdaLoRA. We sincerely refer you to the plots in Figure 5.
>
> There are some similar features. 1)  A lot of singular values are nearly zero. AdaLoRA gradually prunes ranks to make some of the singular values zero. BiLoRA utilizes bi-level optimization to automatically learn this distribution. 2) There are some important ranks with large singular values.
>
> More results can be analyzed in future work in the distribution of singular values of optimal solutions. We expect BiLoRA can provide flexible interaction with singular values and help us better understand the process of low-rank fine-tuning.

---

> ### Author Response · Authors · 2023-11-22
>
> Dear Reviewer,
>
> Thank you for your valuable insights and suggestions. We have tried our best to answer your questions in our author response, and we are also working to revise the paper following your suggestions. Given that the discussion period is ending soon, we were wondering if you could let us know if you have further questions or whether the author response addressed your concerns. We would be delighted to answer any further questions you might have.
>
> Sincerely,
>
> Authors

---

### Official Review · Reviewer_hUsX · 2023-11-01

**Soundness:** 3 good
**Presentation:** 3 good
**Contribution:** 2 fair
**Rating:** 5
**Confidence:** 3

**Summary:**

The author proposes bi-level optimization for the SVD LoRA variant. Specifically, two datasets are needed for the corresponding two level optimization objectives. At one level, the proposed method optimizes P and Q, while at the other level, it optimizes \sigma. Two penalty terms are used to regularize PQ and \sigma. Empirical results show competitive model performance and training speed compared with LoRA.

**Strengths:**

+ The SVD LoRA variant is formulated as two optimization objectives, one for optimizing P and Q, while the other for optimizing \sigma.
+ The formulation is easy to follow.
+ Comprehensive empirical results on NLU and NLG are provided.

**Weaknesses:**

+ It seems that the empirical model performance and training speed are mostly similar as LoRA.
+ It'd be better to see results on LLMs (>10B) in NLG tasks.
+ No theoretical insights are provided for understanding LoRA with bi-level optimization. Specifically, it doesn't seem convincing to replace LoRA with BiLoRA.

**Questions:**

+ For the SVD LoRA variant, it seems to me that P and Q are of similar size as W. How does it achieve parameter-efficient finetuning?
+ How do you decide two datasets D1 and D2?

---

> ### Author Response · Authors · 2023-11-16
> **Thank you for your constructive feedback.**
>
> We appreciate your constructive feedback very much. Since we are not able to submit a PDF, we provide all the figures from Figure 2-7 in the Supplementary Section C-F of our paper. All the indexes of figures are referred to those of the updated paper. We sincerely refer you to the figures in our paper. We provide our response to your questions as follows.
>
> >Weakness 1: It seems that the empirical model performance and training speed are mostly similar as LoRA.
>
> **For performance:**
>
> For NLU tasks, our method, BiLoRA, has demonstrated significant performance improvements over baselines on multiple datasets and models. As shown in Table 1 in the paper, using BiLoRA to fine-tune the RoBERTa model with different-sizes on the eight datasets of GLUE surpassed other baselines on almost all datasets. The average performance is significantly higher than other baselines, with 0.7, 0.8, 0.9 margin respectively.. On the DeBERTa-v3 model, AdaLoRA outperformed LoRA by 0.7 on average across the eight datasets. Our method, surpassing LoRA by 1.3, achieved a remarkably superior performance over LoRA and other baselines.
> To further demonstrate the performance improvement of BiLoRA, we further conduct experiments on GPT-2 on 2 more NLG datasets, DART and WebNLG.
>
> | BLEU           | WebNLG | DART |
> |----------------|--------|------|
> | Full Finetune  | 46.5   | 46.2 |
> | Adapter L      | 50.2   | 42.4 |
> | FTtop2         | 54.9   | 41.0 |
> | Prefix         | 36.0   | 46.4 |
> | LoRA           | 55.3   | 47.1 |
> | BiLoRA         | **56.1**   | **49.0** |
>
> To the best of our knowledge, the results of fine-tuning RoBERTa, DeBERTa and GPT-2 on these NLU and NLG tasks of BiLoRA are the best among all the existing fine-tuning methods. Results on different models and on both NLU and NLG datasets show that our method, BiLoRA outperforms other baselines by a large margin, demonstrating the effectiveness of our method.
>
> **For speed:**
>
> Firstly, we provide the total training steps needed for convergence for LoRA and BiLoRA in the following tablet. We use the results of  RoBERTa-base on MNLI and SST2 datasets.
> | Method/steps   | MNLI       | SST2       |
> |----------------|------------|------------|
> | Ro(LoRA)       | 184k       | 63k        |
> | Ro(BiLoRA)     | 2*15k(1/6×)  | 2*3k(1/11×)  |
>
> Secondly, we provide the total training time training steps needed for convergence for AdaLoRA and BiLoRA in the following tablet. We use the results of  DeBERTa-v3-base on MNLI and SST2 datasets. The time is measured in min.
> | Method/Time   | MNLI            | SST2           |
> |---------------|-----------------|----------------|
> | Dv3(AdaLoRA)     | 753.54          | 240.57         |
> | Dv3(BiLoRA)   | 446.21(1/1.7×)  | 56.71(1/4.2×)  |
> |               |                 |                |
> | **Method/Steps**  | **MNLI**            | **SST2**           |
> | Dv3(AdaLoRA)     | 85.9k           | 50.5k          |
> | Dv3(BiLoRA)   | 2*15k(3×)       | 2*3k(8×)       |
>
> Results show that BiLoRA uses 6 times/11 times less steps than LoRA for convergence and  that BiLoRA uses 3 times/8 times less steps than AdaLoRA for convergence.
> Results demonstrate that BiLoRA can converge much faster than LoRA, AdaLoRA and take much less time for training than baselines. BiLoRA is especially time-efficient on larger datasets such as MNLI, which can reduce the total time of LoRA from 3191 min to 1407 min.
>
> >Weakness 2: It'd be better to see results on LLMs (>10B) in NLG tasks.
>
> Due to constraints of time and GPU resources, we are not able to conduct experiments on very large-scale LLMs. To demonstrate the effectiveness of BiLoRA when scaling up, we further conduct experiments on DeBERTa-xxlarge(1.5B) on 2 more datasets (SST2, QNLI) to demonstrate the effectiveness of BiLoRA.
>
> | Methods/Acc | SST2 | QNLI |
> |-------------|------|------|
> | LoRA        | 96.9 | 96.0 |
> | BiLoRA      | **97.3** | **96.5** |
>
> Results demonstrate the effectiveness of BiLoRA when scaling up to large language models.

---

> ### Author Response · Authors · 2023-11-16
> **Response to other constructive feedback and questions.**
>
> >Weakness 3: It doesn't seem convincing to replace LoRA with BiLoRA due to the lack of theoretical insights.
>
> We’d like to offer some empirical insights and notable advantages of BiLoRA in 2 aspects.
> * Limitations of existing fine-tuning methods.
> * Inspirations and advantages of bi-level optimization.
>
> First, existing fine-tuning methods can suffer from overfitting. The large number of parameters in pre-trained models may make the full fine-tuning process more prone to overfitting[1]. Existing PEFT methods do not well solve this problem. The generalization capability of a fine-tuned model on the evaluation dataset can reflect the range of overfitting. We plot the training loss and evaluation loss of LoRA, AdaLoRA and BiLoRA to illustrate the generalization capability. We conduct experiments on RoBERTa-base on CoLA dataset and clip the first 2k steps. Figure 2(a) and Figure 2(b) show that for both LoRA and AdaLoRA, the evaluation loss starts to increase at or before 1k step while the training loss still decreases. The gap between the 2 losses is getting huge. This demonstrates the limitations of existing methods that they still suffer from overfitting issues. Figure 2(c) shows that both inner training loss and outer training loss are close to the evaluation loss. The evaluation loss is still decreasing after 2k steps and is obviously lower than that of LoRA and AdaLoRA. These results convincingly demonstrate the limitations of existing methods and the effectiveness of BiLoRA in addressing overfitting.
>
> Second, we gain inspiration from DARTS (Differentiable Architecture Search) which utilizes a bi-level optimization for searching network architectures. DARTS uses training dataset and validation dataset separately for training operations and their corresponding proportions in 2 levels. DARTS conducts experiments on comparison between bi-level optimization and the normal training method(single-level). The latter performs much worse and they concludes that the latter “would cause α (analogous to hyperparameters) to overfit the training data, leading to poor generalization”[2]. We gain this inspiration of preventing overfitting and innovatively explore the possibilities of applying bi-level optimization to PEFT domain for addressing the existing limitations of overfitting. Extensive experiments demonstrate the effectiveness of BiLoRA.
>
> >Question 1: How does it achieve parameter-efficient finetuning?
>
> In LoRA, for $h=W_{0}x$, the modified forward pass yields:
> $h = W_{0}x + \Delta Wx = W_{0}x + BAx$,
> where $\Delta W \in R^{d \times k}$, $A \in R^{r \times d}$, $B \in R^{k \times r}$ and $r \ll min\{d,k\}$. So the total number of trainable parameters is much smaller than full fine-tuning. In BiLoRA, P is of the same size as B, Q is of the same size as A and the singular value matrix $\Lambda \in R^{r \times r}$, which is much smaller than A and B. So the total count of the trainable parameters is almost the same as LoRA and much less than full fine-tuning. So this can effectively achieve parameter-efficient finetuning. In addition, BiLoRA accelerates the training process, thus making PEFT more efficient.
>
> >Question 2: How do you decide two datasets D1 and D2?
>
> The Dataset Partition, together with learning rate can help keep the balance of inner/outer optimization, which can contribute greatly to preventing the model from overfitting. Lower level has more trainable parameters, so it’s natural to use more data for training singular vectors, while using the left for training singular values. We further experiment on DeBERTa-v3-base on CoLA and SST2 datasets to show the influence of different dataset partitions. We change the inner level dataset partition from 0.6 to 1.0 with 0.1 interval.
> | Data Partition | CoLA  | SST2  |
> |----------------|-------|-------|
> | 0.6            | 68.01 | 94.84 |
> | 0.7            | 70.16 | 95.87 |
> | 0.8            | **70.52** | 96.02 |
> | 0.9            | 70.39 | **96.44** |
> | 1.0            | 67.94 | 94.51 |
>
> Results show that too small partitions(<=0.6) or too large partitions(1.0, only train singular matrix) can harm the overall performance.When the inner partition is too small, singular vectors are not well trained and when the inner partition is 1.0, singular vectors are not trained at all, which can cause a great performance drop. These results can also show that the bi-level optimization is efficient and the two levels are both necessary in preventing overfitting and enhancing performances.
> In the paper, we don’t ever change the partition of the data and keep it 8:2. Tuning the partition may further improve the overall performance.
>
> [1] Rabeeh Karimi Mahabadi, James Henderson, and Sebastian Ruder. Compacter: Efficient low-rank hypercomplex adapter layers.
>
> [2] Liu H, Simonyan K, Yang Y. Darts: Differentiable architecture search.

---

> ### Author Response · Authors · 2023-11-22
>
> Dear Reviewer,
>
> Thank you for your valuable insights and suggestions. We have tried our best to answer your questions in our author response, and we are also working to revise the paper following your suggestions. Given that the discussion period is ending soon, we were wondering if you could let us know if you have further questions or whether the author response addressed your concerns. We would be delighted to answer any further questions you might have.
>
> Sincerely,
>
> Authors

---

### Official Review · Reviewer_YcPD · 2023-11-05

**Soundness:** 3 good
**Presentation:** 4 excellent
**Contribution:** 2 fair
**Rating:** 6
**Confidence:** 3

**Summary:**

Parameter-efficient fine-tuning (PEFT) is an important line of work for adapting large foundation models to a specific task. LoRA is one of the promising approaches in this direction which models the update matrices as product of two low-rank matrices. A recent approach AdaLoRA further refines LoRA formulation by modeling the update matrices as low-rank pseudo SVD decomposition and uses the singular values to adaptively allocate parameter budget to different matrices. This paper proposes a bi-level optimization approach for learning the AdaLoRA updates which allows for better generalization. The results show consistent improvement over LoRA across different tasks and models.

**Strengths:**

- The paper is well-written and easy to follow
- Results show consistent gains over LoRA with reduced training times due to faster convergence
- Although, the bi-level optimization approach is not novel in itself but its application in context of PEFT is new

**Weaknesses:**

- Motivation for using bi-level optimization is not well supported, more specifically "One limitation of AdaLoRA is that it learns pseudo singular vectors in {P, Q} and pseudo singular values in Λ simultaneously by minimizing the fine-tuning loss on a single training dataset", why is this necessarily a limitation?
- The approach is compared against AdaLoRA only in one of the experiments since it is the most relevant baseline for this paper so I'd expected a more thorough comparison against AdaLoRA

**Questions:**

- Does BiLoRA also apply iterative pruning of singular values (similar to AdaLoRA)? (the paper seems to talk only about the optimization of pseudo-SVD matrices)
- Table 7 compares the total training time between LoRA and BiLoRA and shows BiLoRA is faster due to faster convergence, what are the total training steps needed for convergence for these two methods? what are the per-update cost differences between these two? Also, can we do the cost comparison with AdaLoRA?
- In the approximately binary parameterization of singular values, does removing the regularization help i.e. just sigmoid over the real values  (since this would be similar to real value baseline but with only positive values)?

---

> ### Author Response · Authors · 2023-11-16
> **Thank you for your constructive feedback.**
>
> We appreciate your constructive feedback very much. Since we are not able to submit a PDF, we provide all the figures from Figure 2-7 in the Supplementary Section C-F of our paper. All the indexes of figures are referred to those of the updated paper. We sincerely refer you to the figures in our paper. We provide our response to your questions as follows.
> >Weakness 1: Motivation is not well supported.
>
> The motivation of BiLoRA can be concluded in 4 aspects.
> * Limitations of existing fine-tuning methods.
> * Inspirations and advantages of bi-level optimization.
> * Accelerating the fine-tuning process.
> * Opportunities for deeper understanding of PEFT process.
>
> First, existing fine-tuning methods can suffer from overfitting. The large number of parameters in pre-trained models may make the full fine-tuning process more prone to overfitting[1]. Existing PEFT methods do not well solve this problem. The generalization capability of a fine-tuned model on the evaluation dataset can reflect the range of overfitting. We plot the training loss and evaluation loss of LoRA, AdaLoRA and BiLoRA to illustrate the generalization capability. We conduct experiments on RoBERTa-base on CoLA dataset and clip the first 2k steps. Figure 2(a) and Figure 2(b) show that for both LoRA and AdaLoRA, the evaluation loss starts to increase at or before 1k step while the training loss still decreases. The gap between the 2 losses is getting huge. This demonstrates the limitations of existing methods that they still suffer from overfitting issues. Figure 2(c) shows that both inner training loss and outer training loss are close to the evaluation loss. The evaluation loss is still decreasing after 2k steps and is obviously lower than that of LoRA and AdaLoRA. These results convincingly demonstrate the limitations of existing methods and the effectiveness of BiLoRA in addressing overfitting.
>
> Second, we gain inspiration from DARTS (Differentiable Architecture Search) which utilizes a bi-level optimization for searching network architectures. DARTS uses training dataset and validation dataset separately for training operations and their corresponding proportions in 2 levels. DARTS conducts experiments on comparison between bi-level optimization and the normal training method(single-level). The latter performs much worse and they concludes that the latter “would cause α (analogous to hyperparameters) to overfit the training data, leading to poor generalization”[2]. We gain this inspiration of preventing overfitting and innovatively explore the possibilities of applying bi-level optimization to PEFT domain for addressing the existing limitations of overfitting. Extensive experiments demonstrate the effectiveness of BiLoRA.
>
> Third, one important goal of PEFT is to accelerate the fine-tuning process. The bi-level optimization framework effectively accommodates larger learning rates by iteratively optimizing between the two levels without affecting the training stability. Through BiLoRA, we significantly reduce the training steps and training time needed for convergence, thus making PEFT more efficient.
>
> Fourth, we expect BiLoRA to encourage deeper understanding of low-rank fine-tuning processes and more design potentials. We offer more flexible and direct interaction with singular values and singular matrices at separate levels and design different parameterization of singular values. More efficient methods and deeper insights can be inspired by this bi-level optimization framework.
>
> >Weakness 2: A more thorough comparison against AdaLoRA.
>
> In the paper, we compare BiLoRA with AdaLoRA on DeBERTa-v3. To have a more thorough comparison with AdaLoRA, we further conduct experiments on RoBERTa-base on 4 NLU tasks and GPT-2 on 2 NLG tasks. The experiment results can be seen as below, higher is better for all the scores.
> First, we compare AdaLoRA with BiLoRA on 4 NLU datasets.
> | Method   | MNLI | CoLA | QNLI | QQP | Avg. |
> |----------|------|------|------|-----|------|
> | AdaLoRA  | 87.2 | 63.4 | 92.8 | 91.0 | 83.6 |
> | BiLoRA   | **87.9** | **64.8** | **93.3** | **91.4** | **84.4** |
>
> Second, we compare AdaLoRA on 2 NLG datasets, WebNLG and DART. Results of other methods are from LoRA as a score reference.
>
> | BLEU           | WebNLG | DART |
> |----------------|--------|------|
> | Full Finetune  | 46.5   | 46.2 |
> | Adapter L      | 50.2   | 42.4 |
> | FTtop2         | 54.9   | 41.0 |
> | Prefix         | 36.0   | 46.4 |
> | LoRA           | 55.3   | 47.1 |
> | AdaLoRA        | 55.5   | 47.6 |
> | BiLoRA         | **56.1**   | **49.0** |
>
> Results on different models and on both NLU and NLG datasets show that our method, BiLoRA outperforms LoRA, AdaLoRA and other baselines by a large margin, demonstrating the effectiveness of our method.
>
> [1] Rabeeh Karimi Mahabadi, James Henderson, and Sebastian Ruder. Compacter: Efficient low-rank hypercomplex adapter layers.
>
> [2] Liu H, Simonyan K, Yang Y. Darts: Differentiable architecture search.

---

> ### Author Response · Authors · 2023-11-16
> **Response to your constructive questions.**
>
> >Question 1:Does BiLoRA also apply iterative pruning of singular values (similar to AdaLoRA)?
>
> In the paper, BiLoRA doesn’t prune singular values. In fact, the pruning process can be applied easily to BiLoRA with only several code lines since singular values are separately in the upper level and can be interacted with directly.
> We expect BiLoRA to be powerful and effective, yet easy to use. The number of hyperparameters for BiLoRA is kept nearly the same as LoRA, which is much less than AdaLoRA. For such simplicity, we don’t apply pruning which will introduce some extra complexities.
> After all, BiLoRA is highly compatible with existing low-rank adaptation methods and we are looking forward to combining other LoRA methods with BiLoRA to achieve higher efficiency and better performance.
>
> >Question 2: Total training steps and per-update cost for LoRA, BiLoRA and AdaLoRA.
>
> Firstly, we provide the total training steps needed for convergence for LoRA and BiLoRA and the per-update cost for the two methods in the following tablet. We use the results of  RoBERTa-base on MNLI and SST2 datasets. The per-update cost is measured in min/k.
> | Method/steps   | MNLI       | SST2       |
> |----------------|------------|------------|
> | Ro(LoRA)       | 184k       | 63k        |
> | Ro(BiLoRA)     | 2*15k(1/6×)  | 2*3k(1/11×)  |
> |                |            |            |
> | **Method/per-update** | **MNLI**     | **SST2**       |
> | Ro(LoRA)       | 17.34      | 17.40      |
> | Ro(BiLoRA)     | 46.90(2.7×)| 43.33(2.49×)|
>
> Results show that BiLoRA uses 6 times/11 times less steps than LoRA for convergence. The per-step cost for BiLoRA is roughly 2.5 times as LoRA, since BiLoRA needs to iteratively optimize between the two levels and the calculation of outer hypergradients can cost more than simple gradient calculation. Results demonstrate that BiLoRA can converge much faster than LoRA and takes much less time for training than LoRA.
>
> Secondly, we provide the total training steps needed for convergence for AdaLoRA and BiLoRA and the per-update cost for the two methods in the following tablet. We use the results of  DeBERTa-v3-base on MNLI and SST2 datasets. According to AdaLoRA paper, it can be seen that each training epoch of AdaLoRA is longer than LoRA, thus intuitively BiLoRA is also faster than AdaLoRA.   The per-update cost is measured in min/k and time is measured in min.
>
> | Method/Time   | MNLI            | SST2           |
> |---------------|-----------------|----------------|
> | Dv3(AdaLoRA)     | 753.54          | 240.57         |
> | Dv3(BiLoRA)   | 446.21(1/1.7×)  | 56.71(1/4.2×)  |
> |               |                 |                |
> | **Method/Steps**  | **MNLI**            | **SST2**           |
> | Dv3(AdaLoRA)     | 85.9k           | 50.5k          |
> | Dv3(BiLoRA)   | 2*15k(1/3×)       | 2*3k(1/8×)       |
> |               |                 |                |
> | **Method/per-update** | **MNLI**         | **SST2**           |
> | Dv3(AdaLoRA)     | 8.77            | 4.76           |
> | Dv3(BiLoRA)   | 14.87(1.7×)     | 9.45(2.0×)     |
>
> Results show that BiLoRA uses 3 times/8 times less steps than AdaLoRA for convergence. The per-step cost for BiLoRA is roughly 1.7/2.0 times as AdaLoRA. Results demonstrate that BiLoRA can converge much faster than LoRA, AdaLoRA and take much less time for training than baselines.
>
> >Question 3: Does removing the regularization, such as just sigmoiding over the real values help?
>
> It’s a really good suggestion of just using sigmoid over the real values without regularization. We further conduct experiments on RoBERTa-base on MNLI, SST2 and MRPC datasets.
>
> | Method/Acc     | MNLI | SST2 | MRPC |
> |----------------|------|------|------|
> | Softmax        | **87.9** | **95.1** | 91.7 |
> | Reg_Binary     | 87.6 | 94.8 | 91.4 |
> | Binary         | 87.8 | 94.7 | **92.1** |
>
> Results show that using sigmoid over the real values without regularization can achieve equal or better performance than with regularization and achieve better performance than Softmax on MRPC dataset.
> The design purpose of Binary with regularization is to automatically assign either a high or low importance to each singular vector pair with the corresponding singular value as zero or one. Since singular values can be seen as importance scores of different ranks, the ranks with zero singular values can be pruned. The whole method can serve as an automatic rank selection mechanism. In the future, we will further explore the potentials of this design and improve the performance.

---

> > ### Comment · Reviewer_YcPD · 2023-11-21
> >
> > Thanks for the detailed response, although I am still not sure about the need for bi-level optimization here, the empirical evidence provided by authors shows consistent improvements with BiLoRA. I am improving my rating to 6.

---

> ### Author Response · Authors · 2023-11-22
>
> Thank you for all the constructive feedback. We really appreciate your insightful suggestions.
>
> We highly value your concerns regarding the motivation, and we genuinely provide the following response in the hope of alleviating your concerns.
>
> As far as we are concerned, the reason for utilizing bi-level optimization for low-rank adaptation is mainly from 3 aspects.
> * Convincing results and inspirations from DARTS. DARTS conducts experiments on comparison between bi-level optimization and the normal training method(single-level). Results demonstrate that the latter performs much worse and DARTS concludes that the latter “would cause α (analogous to hyperparameters) to overfit the training data, leading to poor generalization”.
> * The nature of bi-level optimization. Iteratively optimizing between the 2 levels can intuitively prevent overfitting on either of the 2 subsets.
> * The overfitting issue has long been a hindrance to the further improvement of existing fine-tuning methods.
>
> Extensive experiment results demonstrate the better generalization capability of BiLoRA and loss curves convincingly illustrate that BiLoRA largely addresses the overfitting issue.
>
> BiLoRA innovatively introduces bi-level optimization to the Parameter-Efficient-Fine-Tuning domain. It explores and demonstrates the huge potentials of utilizing bi-level optimization for improving the generalization capability of low-rank adapters and speeding up the fine-tuning process. We trust BiLoRA will help enhance the efficiency and performance of many models on many downstream tasks.
>
> Thank you again for engaging with us and your valuable feedback.

---

### Author Response · Authors · 2023-11-21

Dear Reviewers,

Thank you for your valuable insights and suggestions. We have tried our best to answer your questions in our author response, and we are also working to revise the paper following your suggestions. Given that the discussion period is ending soon, we were wondering if you could let us know if you have further questions or whether the author response addressed your concerns. We would be delighted to answer any further questions you might have.

Sincerely,

Authors

---

### Author Response · Authors · 2023-11-22

Thank you so much for your insightful suggestions. We sincerely appreciate your helpful feedback.

According to all the valuable reviews, the main concern of BiLoRA concentrates on the motivation.
As far as we are concerned, the reason for utilizing bi-level optimization for low-rank adaptation is mainly from 3 aspects.
* Convincing results and inspirations from DARTS. DARTS conducts experiments on comparison between bi-level optimization and the normal training method(single-level). Results demonstrate that the latter performs much worse and DARTS concludes that the latter “would cause α (analogous to hyperparameters) to overfit the training data, leading to poor generalization”.
* The nature of bi-level optimization. Iteratively optimizing between the 2 levels can intuitively prevent overfitting on either of the 2 subsets.
* The overfitting issue has long been a hindrance to the further improvement of existing fine-tuning methods.

Extensive experiment results demonstrate the better generalization of BiLoRA. Loss curves also illustrate that BiLoRA largely addresses the overfitting issue.

BiLoRA innovatively introduces bi-level optimization to the Parameter-Efficient-Fine-Tuning domain. It explores and demonstrates the huge potentials of utilizing bi-level optimization for improving the generalization capability of low-rank adapters and speeding up the fine-tuning process. We trust BiLoRA will help enhance the efficiency and performance of many models on many downstream tasks.

---

### Author Response · Authors · 2023-11-23

Dear Reviewers,

Thank you for all the constructive feedback. We really appreciate your insightful suggestions.

We highly value your concerns regarding the motivation and the reason why techniques work well, and we genuinely provide the following response in the hope of alleviating your concerns.

Extensive experiment results demonstrate the better generalization capability of BiLoRA and loss curves convincingly illustrate that BiLoRA largely addresses the overfitting issue.
There may be 3 main useful techniques for the improvement. 1) Bi-level optimization framework. 2) Parameterization of Singular Values. 3) Orthogonality Initialization and Regularization.

**Bi-level Optimization Framework.**

As far as we are concerned, the reason for utilizing bi-level optimization for low-rank adaptation and the powerful effect is mainly from 3 aspects:
* The overfitting issue has long been a hindrance to the further improvement of existing fine-tuning methods.
* Convincing results, conclusions and inspirations from DARTS. DARTS conducts experiments on comparison between bi-level optimization and the normal training method(single-level). Results demonstrate that the latter performs much worse and DARTS concludes that the latter “would cause α (analogous to hyperparameters) to overfit the training data, leading to poor generalization”. Bi-level optimization has been shown to improve the generalization capability.
* The nature of bi-level optimization. Iteratively optimizing between the 2 levels can intuitively prevent overfitting on either of the 2 subsets.

**Parameterization of Singular Values.**

The parameterization of singular values can help improve the performance because:
*  Softmax and Approximately Binary Parameterization outperform Real Value (the parameterization in LoRA and AdaLoRA) because they yield positive values which meet the constraint that singular values need to be non-negative while Real Value does not.

**Orthogonality Initialization and Regularization.**

The Orthogonality Initialization and Regularization can help improve the performance due to:
* With $P$ and $Q$ to be almost orthogonal, $\Lambda$ can be exactly the singular value matrix of the low-rank adapters according to SVD. Thus the outer level is directly optimizing the singular values which can be viewed as importance of diifferent ranks.
* The Orthogonality Regularization can contribute to preventing overfitting since it helps optimizing the training loss while still keeping the evaluation loss decreasing rather than overfitting. This has been convincingly illustrated by the loss curves.


BiLoRA innovatively introduces bi-level optimization to the Parameter-Efficient-Fine-Tuning domain. It explores and demonstrates the huge potentials of utilizing bi-level optimization for improving the generalization capability of low-rank adapters and speeding up the fine-tuning process. We trust BiLoRA will help enhance the efficiency and performance of many models on many downstream tasks.

Thank you again for engaging with us and your valuable feedback.

---

### Meta-Review · Area_Chair_UMHy · 2023-12-07

**Metareview:**

The paper provides a bi-level optimization scheme for parameter efficient fine tuning (PEFT) of LLMs. The paper improves upon a recent approach named AdaLoRA. The use of bi-level optimization is novel to PEFT and the experiments show some promising results. Furthermore, the reviews appreciated some observations like the use of soft-max for singular values as a method to make sure they remain positive. The main concern pointed out a lack of sufficient justification for using a bi-level optimization scheme, namely analysis showing why it helps, e.g. by detecting specific failure types that are avoided with the bi-level approach. Given that this is the main contribution of the paper, I see this issue as a major one. My conclusion from the reviews and discussion is that the paper requires more work, specifically mitigating the mentioned gap before it can be published in a venue as ICLR.

**Justification For Why Not Higher Score:**

the main contribution is a bi-level optimization scheme but the analysis for why it improves upon a direct optimization method is not convincing enough

**Justification For Why Not Lower Score:**

n/a

---

### Decision · Program_Chairs · 2024-01-16

Reject